# NON-EQUILIBRIUM DYNAMICS OF HYBRID CONTINUOUS-DISCRETE GROUND-STATE SAMPLING

**Timothée Guillaume Leleu**
NTT Research, Stanford University
timothee.leleu@ntt-research.com
tleleu@stanford.edu

**Sam Reifenstein**
NTT Research
samuel.reifenstein@ntt-research.com

## ABSTRACT

We propose a general framework for a hybrid continuous-discrete algorithm that integrates continuous-time deterministic dynamics with Metropolis-Hastings (MH) steps to combine search dynamics that either preserve or break detailed balance. Our purpose is to study the non-equilibrium dynamics that leads to the ground state of rugged energy landscapes in this general setting. Our results show that MH-driven dynamics reach "easy" ground states more quickly, indicating a stronger bias toward these solutions in algorithms using reversible transition probabilities. To validate this, we construct a set of Ising problem instances with a controllable bias in the energy landscape that makes certain degenerate solutions more accessible than others. The constructed hybrid algorithm demonstrates significant improvements in convergence and ground-state sampling accuracy, achieving a 100x speedup on GPU compared to simulated annealing, making it well-suited for large-scale applications.

## 1 INTRODUCTION

The fact that non-equilibrium relaxation dynamics reach different degenerate ground states at different rates, contrary to what is expected at equilibrium, is a fundamental topic in optimization and statistical physics (Bernaschi et al., 2020; Lucas, 2014; Mohseni et al., 2022). In machine learning, recent works suggest that sampling from more easily accessible degenerate ground states favors finding models that generalize well during the training of overparameterized neural networks (Soudry et al., 2018; Baity-Jesi et al., 2018; Feng & Tu, 2021; Baldassi et al., 2022; 2023). Here, we study this issue from the general perspective of non-convex optimization, focusing on the Ising Hamiltonian. In particular, we study the task of sampling from degenerate ground states of a non-convex energy landscape. Ground-state sampling involves finding not just any ground state, as it is often defined in combinatorial optimization, but multiple degenerate ground states. In this context, non-equilibrium dynamics refers to the processes through which systems evolve over time toward steady states, with varying rates of convergence to distinct ground states that deviate from equilibrium expectations.

The Metropolis-Hastings (MH) algorithm is one of the most widely used techniques for sampling from a reversible Markov Chain Monte Carlo (MCMC) (Zhang et al., 2020; Chib & Greenberg, 1995; Chen et al., 2014). This approach proposes new states and accepts them with a probability designed to ensure that the stationary distribution of the chain is the Boltzmann distribution. Despite its simplicity and versatility, MH can struggle with getting trapped in local energy minima, especially when dealing with complex energy landscapes. Examples of these energy landscapes often arise in systems exhibiting broken ergodicity (Bernaschi et al., 2020), such as spin glasses. In these cases, other MCMC algorithms that maintain detailed balance, like simulated annealing (Kirkpatrick et al., 1983) and more advanced methods (Hukushima & Nemoto, 1996; Surjanovic et al., 2022; Hukushima & Iba, 2003; Houdayer, 2001; Landau et al., 2004), are often favored for exploring rugged energy landscapes and finding optimal solutions. Interestingly, several algorithms do not satisfy detailed balance yet remain competitive (Leleu et al., 2019; 2021; Goto et al., 2021; Buesing et al., 2011), particularly those based on chaotic search (Leleu et al., 2019; 2021). Although these approaches do not sample in general from the exact Boltzmann distribution at equilibrium, it can be shown in special cases that non-reversible Markov chains mix faster than reversible ones (Kapfer & Krauth,

2017). However, their ability to reach degenerate ground states of a rugged energy landscape using non-equilibrium dynamics is still not well understood.

In this work, we introduce a generalized algorithm that combines MCMC techniques with chaotic search dynamics via the chaotic amplitude control (CAC) algorithm. Tuning the hyperparameters of this hybrid approach allow for a smooth transition between the two strategies. Additionally, we create new planted instances with degenerate ground states, inspired by the Wishart planted models. A tunable parameter controls the bias towards different degenerate ground states, enabling us to investigate the non-equilibrium behavior of the hybrid MH/chaotic algorithm in locating solutions that vary in difficulty. We demonstrate that the algorithm's performance can be optimized based on the bias parameter, making it competitive with state-of-the-art methods. The code for our algorithm can be found on this online repository: GitHub Repository.

## 2 RELATED WORK

### 2.1 DISCRETE STATE MCMC OPTIMIZERS

MCMC methods for optimizing energy functions over discrete spaces—such as simulated annealing (Kirkpatrick et al., 1983) and parallel tempering (Hukushima & Nemoto, 1996)—are standard approaches in the field of optimization. Building upon these, recent schemes such as population annealing (Hukushima & Iba, 2003), variational parallel tempering (Surjanovic et al., 2022) and specialized techniques, including cluster updates (Houdayer, 2001), Wang-Landau sampling (Landau et al., 2004), and Replica Exchange Wang-Landau Sampling (Vogel et al., 2013), have been developed to enhance convergence and sampling efficiency. However, these methods often struggle in higher-dimensional discrete spaces due to the complexity of the energy landscape and the tendency to become trapped in local energy minima caused by broken ergodicity. This is a behavior observed in many systems with frustrated interactions. (Bernaschi et al., 2020).

### 2.2 HYBRID CONTINUOUS-DISCRETE SAMPLERS

For continuous space sampling, a plethora of methods have been developed based on the idea of combining continuous-time dynamics (typically via ODEs or SDEs) with discrete MCMC corrections. Some examples of this are Metropolis Adjusted Langevin Dynamics (MALA) (Roberts & Rosenthal, 1998), Hamiltonian Monte Carlo (HMC) (Neal et al., 2011; Xifara et al., 2014), as well as derivative methods such as the No-U-Turn Sampler (NUTS) (Hoffman et al., 2014), Generalized HMC (Chen et al., 2014), HMC with the Gaussian integral trick (Zhang et al., 2012), amortized Metropolis adjustments (Zhang et al., 2020), and sampling using collective variables (Schönle et al., 2024). These methods primarily target continuous spaces, and thus are not directly applicable to discrete-state sampling. However, research has been conducted to extend continuous sampling techniques to discrete spaces, including methods like discontinuous HMC (Nishimura et al., 2020), Gibbs with Gradient (GWG) (Grathwohl et al., 2021), and the Hamming Ball Sampler (Titsias & Yau, 2017). Nonetheless, these approaches are not suitable for ground-state sampling.

### 2.3 HYBRID CONTINUOUS-DISCRETE OPTIMIZERS

Recently, there has been renewed interest in using continuous-space ODEs to find ground states in discrete spaces. This effort has been driven primarily by the idea of leveraging a diverse array of physical hardware, including CMOS, memristors, spintronics, superconducting circuits, and photonics (see review (Mohseni et al., 2022)), all designed to solve optimization problems more efficiently than current high-performance digital computers.

In particular, dynamical systems such as simulated bifurcation machines (Goto et al., 2021), analog iterative machines (AIM) (Kalinin et al., 2023), and chaotic amplitude control (CAC) (Leleu et al., 2019; 2021; Reifenstein et al., 2023) offer an alternative approach. Rather than relying on gradient descent of an energy function in systems with symmetric interactions, such as in the seminal work by Hopfield (Hopfield & Tank, 1985), it has been shown that Hamiltonian dynamics (Goto et al., 2021; Neal et al., 2011) or chaotic dynamics driven by asymmetric interactions (Leleu et al., 2019; 2021) can outperform standard heuristics like simulated annealing (Kirkpatrick et al., 1983), parallel tempering (Hukushima & Iba, 2003), and breakout local search (Benlic & Hao, 2013) on specific

benchmark datasets (Leleu et al., 2021). This has challenged the prevailing belief that MCMC heuristics maintain an advantage in discrete optimization over systems whose dynamics are relaxed into the continuous domain.

Hybrid approaches that combine continuous-space ODEs with discrete MCMC probabilistic transitions remain underexplored in the context of ground-state search, as opposed to their more common application in Boltzmann sampling.

## 3 METHOD

### 3.1 OVERVIEW

In this section, we introduce a novel MCMC method that combines chaotic search with the MH algorithm (Hastings, 1970) (see Fig. 1 a). The algorithm is built on three main concepts: (1) relaxation from discrete to continuous space (or continuous embedding), (2) addition of auxiliary variables to escape from local minima (chaotic amplitude control (Leleu et al., 2019; 2021) and momentum (Kalinin et al., 2023)), (3) probabilistic jumps in the discrete space based on the Metropolis-Hastings criterion.

In the next section, we construct a novel set of Ising problem instances inspired from the Wishart planted ensemble (Hamze et al., 2020) in order to benchmark this algorithm's ability to sample from optimal solutions (or ground-state energies). The constructed instances exhibit degenerate ground states that are sampled with unequal probabilities by local search algorithms (see Fig. 1 b).

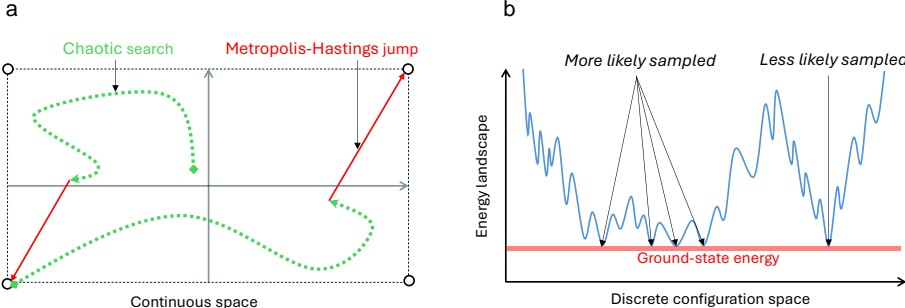

Figure 1: a) Proposed hybrid approach generates deterministic chaotic trajectories that rapidly explores the energy landscape and probabilistic jumps accepted via the Metropolis-Hastings criterion. (b) Constructed planted instances with degenerate ground states with unequal sampling probabilities by heuristics. Some ground states are easier, and others harder, for local search heuristics to discover.

### 3.2 DETERMINISTIC CONTINUOUS TIME MODEL

We define a continuous time dynamical system inspired by the CAC algorithm (Leleu et al., 2019; 2021) and generalize its formulation to draw a correspondence between its parameters and the inverse temperature $\beta$ of the Boltzmann distribution $P(\boldsymbol{\sigma}) \propto e^{\beta V(\boldsymbol{\sigma})}$. The goal is to design a dynamical system capable of sampling from the ground states of $V$, specifically from the zero-temperature distribution $P(\boldsymbol{\sigma})$ defined on the discrete space $\boldsymbol{\sigma} \in \{-1, 1\}^N$, where $N$ is the number of variables or spins.

First, discrete variables $\boldsymbol{\sigma}$ are relaxed to the continuous space $\boldsymbol{x} \in \mathbb{R}^N$ and we define the potential (or energy function) $V(\boldsymbol{x})$ on the real space. The following dynamical system is used to direct the search towards lower energy states:

$$\gamma \frac{d^2 \boldsymbol{u}}{dt^2} + \frac{d\boldsymbol{u}}{dt} = -\alpha \boldsymbol{u} - \boldsymbol{e} \circ \nabla_x V, \tag{1}$$

where the symbol $\circ$ denote the Hadamard product; $\nabla_x V$, the gradient of $V$ with respect to the vector $\boldsymbol{x}$; $\alpha$, a positive parameter; and $\boldsymbol{e}$, a vector of positive auxiliary variables. The variables $\boldsymbol{x}$ are often referred to as "soft spins." The variables $\boldsymbol{u}$ represent the internal states of these soft spins, while the auxiliary variable $e$ accounts for variations in their amplitude (Leleu et al., 2019). The term proportional to $\gamma$ represents the momentum which has been utilized for non-convex

optimization (Kalinin et al., 2023) and sampling (Neal et al., 2011; Xifara et al., 2014). Moreover, we set:

$$x_i = \phi(\beta_{\text{eff}} u_i), \forall i \in \{1, \cdots, N\}, \tag{2}$$

and $\phi(x) = \frac{2}{1+e^{-x}} - 1$ is a sigmoidal function normalized to the domain $x \in [-1, 1]$. $\beta_{\text{eff}}$ is an effective temperature (see Appendix section S3.1 for a discussion about its influence).

The auxiliary variables $e_i$ undergo dynamic changes over time, with the goal of rectifying amplitude heterogeneity, given as follows (Leleu et al., 2019):

$$\frac{d\boldsymbol{e}}{dt} = -\xi(\boldsymbol{x} \circ \boldsymbol{x} - a) \circ \boldsymbol{e}, \tag{3}$$

with $a$ and $\xi$ being parameters: $a$ represents target amplitude for the variables $x$ and $\xi$ represents the speed of the error correction dynamics. Because this algorithm adds momentum dynamics to the original chaotic amplitude control equations (Leleu et al., 2019; 2021), we call it chaotic amplitude control with momentum (CACm).

In the Appendix section S1, the role of the auxiliary variables $\boldsymbol{e}$ is analyzed and we show that their modulation can be interpreted as a dynamic pre-conditioning of the Hessian of $V$ at the proximity of the discrete state. Because the auxiliary variables $\boldsymbol{e}$ remain positive, they do not change the inertia (number of positive, negative, and zero eigenvalues) of the Hessian of $V$ but can change its directions and eigenvalues. Thus, CAC performs some dynamic deformation of the potential function $V$ which affects its ability to sample local minima determined by the shape of $V$ when compared to other algorithms, as it will be shown in the following. Note that the momentum and pre-conditioning play distinct roles in the dynamics. The momentum aggregates past gradients, while pre-conditioning modulates the effective curvature of $V$.

### 3.3 PROBABILISTIC DISCRETE TIME MODEL AND MCMC

In order to utilize this dynamical system algorithmically, we discretize eqs. 1-2 using a straightforward Euler approximation. Unlike Hamiltonian Monte Carlo, which requires careful selection of the integration method, an accurate integration scheme is not essential in this context since the goal is to sample from a discrete distribution rather than precisely from a continuous one. The utilization of a simple integration method and large integration time step enables the discrete-time approximation of the system to function as computationally efficient solver for combinatorial optimization. The discrete-time model is defined as follows (after a suitable change of variables):

$$\boldsymbol{u}(m + 1) = \boldsymbol{u}(m) - \alpha\boldsymbol{u}(m) - \boldsymbol{e}\nabla_x V(\boldsymbol{x}(m)) + \gamma(\boldsymbol{u}(m) - \boldsymbol{u}(m - 1)), \tag{4}$$

$$\boldsymbol{e}(m + 1) = \boldsymbol{e}(m) - \xi(\boldsymbol{x}(m) \circ \boldsymbol{x}(m) - a) \circ \boldsymbol{e}(m). \tag{5}$$

To maintain the effect of the deterministic dynamics when used for optimization, we formulate an MCMC as follows. The eqs. 4 and 5 are iterated for $n$ time steps, subsequent to which we sample a discrete state $\boldsymbol{\sigma} \in \{-1, 1\}^N$ as detailed below:

$$P(\sigma_i = 1) = \frac{x_i + 1}{2}, \sigma_i = -1 \text{ otherwise.} \tag{6}$$

It is worth noting that all spins are updated synchronously and independently. While the deterministic dynamics over $n$ time-steps can be perceived as local updates, the generation of $\boldsymbol{\sigma}$ can leap to remote states. Note moreover that when we set $\xi = 0$, $e_i(0) = 0$, $\alpha = 1$, $m = 1$, and update one spin $i$ at a time, we revert to the equation of the Boltzmann machine.

### 3.4 METROPOLIS ADJUSTED CHAOTIC SAMPLING

When $\xi \neq 0$, detailed balance is violated, causing the stationary distribution of the chain to differ from the Boltzmann distribution. To design an MCMC method that converges (at least in principle) to the Boltzmann distribution, we introduce a Metropolis-Hastings acceptance criterion for the discrete state samples $\sigma(kt)$, where $k$ belongs to $\{1, 2, ..., K\}$. The inverse of the number of Metropolis-Hastings steps is denoted $f_{\text{MH}} = \frac{1}{K}$ and the total number of steps is noted $T$ with $T = Kn$.

Typically, the acceptance criterion of a new configuration $\boldsymbol{\tau}$, when the current state is $\boldsymbol{\sigma}$ is given as follows:

$$A(\boldsymbol{\tau}/\boldsymbol{\sigma}) = \min(1, \frac{P(\boldsymbol{\tau})Q(\boldsymbol{\sigma}/\boldsymbol{\tau})}{P(\boldsymbol{\sigma})Q(\boldsymbol{\tau}/\boldsymbol{\sigma})}). \tag{7}$$

Here, $Q(\boldsymbol{\tau}/\boldsymbol{\sigma})$ represents the probability of transitioning from $\boldsymbol{\sigma}$ to $\boldsymbol{\tau}$. The Monte Carlo step for our method is comprised of two substeps as follows: (1) A deterministic trajectory following eqs. 4 and 5 for $n$ steps starting from $\boldsymbol{u}(kn) = \vec{0}$, $\boldsymbol{e}(kn) = \vec{0}$, and , $\boldsymbol{x}(kn) = \boldsymbol{\sigma}(kn)$, (2) A sample $\boldsymbol{\tau}$ generated with the probability $\boldsymbol{p}$ with $\boldsymbol{p} = \frac{\boldsymbol{y}+1}{2}$ with $\boldsymbol{y}$ defined as $y_i = \phi(\frac{\beta_{\text{eff}}u_i((k+1)n)}{\alpha e_i((k+1)n)})$, i.e., $P(\tau_i = 1) = p_i$, $\tau_i = -1$ otherwise.

Note that $\boldsymbol{y}$ depends exclusively on $\boldsymbol{\sigma}$, since it is generated deterministically from it. Because of the synchronous updating of all spins, we can write the following definition for $Q(\boldsymbol{\tau}/\boldsymbol{\sigma})$:

$$\log Q(\boldsymbol{\tau}/\boldsymbol{\sigma}) = \sum_{i=1}^{N} \left[ \frac{1 + \tau_i}{2} \log(p_i) + \frac{1 - \tau_i}{2} \log(1 - p_i) \right]. \tag{8}$$

To calculate the transition probability in the reverse Monte Carlo step from $\boldsymbol{\tau}$ to $\boldsymbol{\sigma}$, we need to apply first the deterministic sub-step (1) from $\boldsymbol{\tau}$ to obtain a candidate end point $\tilde{\boldsymbol{x}}^1$, from which we can calculate the probability to generate the random sample $\boldsymbol{\sigma}$ with probability $\tilde{p} = \frac{\tilde{\boldsymbol{x}}+1}{2}$. The acceptance criterion can then be rewritten as follows:

$$\log A(\boldsymbol{\tau}/\boldsymbol{\sigma}) = \min(0, \beta(V(\boldsymbol{\sigma}) - V(\boldsymbol{\tau}) - \sum_{i=1}^{N} \left[ \frac{1 + \tau_i}{2} \log(p_i) + \frac{1 - \tau_i}{2} \log(1 - p_i) \right]$$
$$+ \sum_{i=1}^{N} \left[ \frac{1 + \sigma_i}{2} \log(\tilde{p}_i) + \frac{1 - \sigma_i}{2} \log(1 - \tilde{p}_i) \right]). \tag{9}$$

This acceptance criterion can be computed efficiently since it only requires the logarithm of terms already computed within the deterministic equations. The discrete state $\boldsymbol{\tau}$ is accepted with probability $A(\boldsymbol{\tau}/\boldsymbol{\sigma})$. The Markov chain using the Metropolis-Hastings step respects detailed balance condition and converges to our target Boltzmann distribution $P(x) = e^{-\beta V(x)}$.

In summary, the pseudo code of our approach is given in section S2 of the Appendix. Note that $\boldsymbol{y}$ is solely determined by $\boldsymbol{\sigma}$. The deterministic path to $\boldsymbol{y}$ originates from $\boldsymbol{\sigma}$ with $\boldsymbol{e} = 1$. Consequently, the transition probability from $\boldsymbol{\sigma}$ to $\boldsymbol{\tau}$ depends only on $\boldsymbol{\sigma}$ (mapped to $\boldsymbol{y}$ by the deterministic flow), preserving the Markov property. The updates of all spins are done in parallel independently, resulting in a product of probabilities. This can be reduced to a sum of log probabilities for more stable numerical simulations as shown in eq. 9. We call the algorithm defined by eqs. 4-9 Metropolis Hastings Chaotic Amplitude Control with momentum (MHCACm).

## 3.5 SUMMARY OF ALGORITHMS

A key property of new the algorithm proposed in this work is that can be interpreted as a generalization of many existing methods which exist as limiting cases of MHCACm. Some examples are: Simulated annealing (Kirkpatrick et al., 1983), Hopfield neural networks (Hopfield & Tank, 1985), analog iterative machines (Kalinin et al., 2023), chaotic amplitude control (Leleu et al., 2019), and chaotic amplitude control with momentum. These limiting cases are summarized in Table 1. The generalized algorithms tend to have more parameters which need to be accurately tuned in order to adequately utilize the dynamics provided by each aspect of the algorithm. When properly tuned to a given class of instances, the performance of the generalized algorithm performs at least as well as any of its constituent parts. Importantly, automatic parameter tuning methods have recently been developed which allow the many parameters to be quickly and accurately tuned (Reifenstein et al., 2024).

---

[1] $\tilde{\boldsymbol{x}}$ is defined similarly to $\boldsymbol{y}$ using the end state of the deterministic path starting from $\boldsymbol{\tau}$ given as $\tilde{x}_i = \phi(\frac{\beta_{\text{eff}}u_i((k+2)n)}{\alpha e_i((k+2)n)})$

Table 1: Limit cases of the proposed algorithm MHCACm. SA: simulated annealing. HNN: Hopfield neural network. AIM: analog iterative machine. CAC: chaotic amplitude control. CACm: chaotic amplitude control with momentum. MHCACm: Metropolis Hastings Chaotic Amplitude Control with momentum.

| model | $f_{\text{MH}}$ | $\gamma$ | $\xi$ | model | $f_{\text{MH}}$ | $\gamma$ | $\xi$ |
|-------|-----------------|----------|-------|-------|-----------------|----------|-------|
| SA    | $= \frac{1}{T}$ | $= 0$    | $= 0$ | CAC   | $= 1$ | $= 0$ | $> 0$ |
| HNN   | $= 1$ | $= 0$ | $= 0$ | CACm  | $= 1$ | $> 0$ | $> 0$ |
| AIM   | $= 1$ | $> 0$ | $= 0$ | MHCACm | $\frac{1}{T} < f_{\text{MH}} < 1$ | $> 0$ | $> 0$ |

## 4 BENCHMARK

### 4.1 DEGENERATE PLANTED WISHART INSTANCES

We construct a set of planted Ising problem instances that exhibit ground states with different tunable properties. The construction of these instances is inspired by the Wishart planted ensemble (WPE) (Hamze et al., 2020). WPE instances are particularly useful for benchmarking due to their tunable hardness parameter $\alpha_{\text{WPE}}$. For certain values of $\alpha_{\text{WPE}}$, the recovery of the planted solution becomes easier or harder. We introduce the degenerate Wishart planted ensemble (dWPE) to allow for planting of multiple degenerate ground states. Additionally, we add a tunable "bias", denoted $b$, to the energy landscape which allows us to make some ground states easier/harder to find than others for heuristic algorithms as depicted in Fig. 1 (b). Thus, we refer to these different ground states as "easy" and "hard" solutions. Details of the construction method are provided in the Appendix section S5.

### 4.2 COMPUTATION TIME TO FIND EASY AND HARD SOLUTIONS

In this section, we evaluate the novel algorithm for finding low energy states of non-convex combinatorial optimization problems. A common metric for evaluating the performance of Ising solvers is the "time to solution" (TTS) which measures the number of steps needed to have 99% probability of finding any ground state (the smaller, the better the algorithm's performance). In case there are $D$ degenerate ground states (i.e., solutions of same energy), there is equal probability of sampling them from the Boltzmann distribution. However, algorithms generally reach a ground state before fully equilibrating to the Boltzmann distribution and, consequently, each degenerate solution can have different probability of being sampled at finite sampling time. We define TTS, $\text{TTS}_{\text{easy}}$, and $\text{TTS}_{\text{hard}}$ as the time to reach any, the easier, and harder to find degenerate solution (of same Ising energy) with 99% probability as follows:

$$\text{TTS} = T\frac{\log(1 - 0.99)}{\log\left(1 - \sum_d P_d\right)}, \text{TTS}_{\text{easy}} = T\frac{\log(1 - 0.99)}{\log\left(1 - \max_d P_d\right)}, \text{TTS}_{\text{hard}} = T\frac{\log(1 - 0.99)}{\log\left(1 - \min_d P_d\right)}, \quad (10)$$

where $P_d$ is the probability to sample the ground state $d$. The algorithms used for comparison are all tuned to nearly optimal parameters by minimizing the TTS. We use a state-of-the-art autotuning method called dynamic anisotropic smoothing (Reifenstein et al., 2024) (DAS) which is particularly well fitted to optimizing the parameters of heuristics for combinatorial optimization (Reifenstein et al., 2024). The use of a automatic parameter tuning also guarantees a fair comparison between algorithms (see Appendix S4). Moreover, all algorithms are compared using the number of "steps" to solution, where the algorithmic complexity of a single step is dominated by the matrix-vector multiplication of size $N \times N^2$.

### 4.3 GLOBAL OPTIMIZATION TO ANY OPTIMAL SOLUTION

For the task of finding any ground state, Fig. 2 (a) shows that MHCACm performs almost as well as CACm and better than AIM, CAC, and SA[3]. In the case of sampling any ground state of biased

---

[2]For SA, a step represents a full sweep of $N$ spin updates which also has complexity $O(N^2)$. The same for PT, but divided by the number of replica.

[3]Results for SA and PT are collected using dwave-neal (dwavesystems, 2024) and pySA (Mandrà & Munoz-Bauza, 2024)

($b = 12$) dWPE problem instances (see section 4.1), MHCACm requires significantly fewer steps than other algorithm to find an optimal solution (see Fig. 2 (b)).

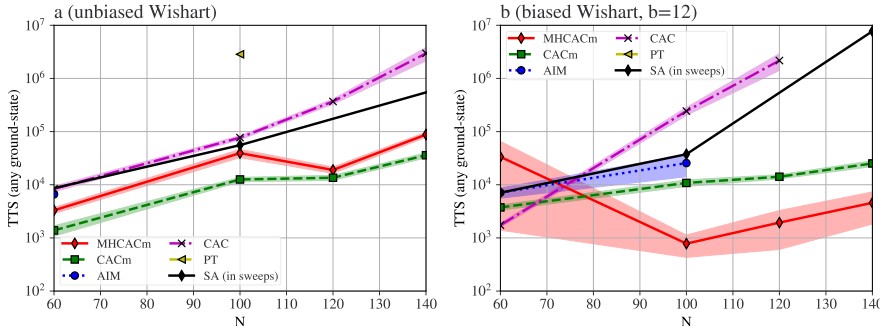

Figure 2: The time to solution of (unbiased) Wishart planted instances and biased degenerate planted instance with $b = 12$ are shown in (a) and (b), respectively. $f_{\text{MH}} = 0.1$. All other parameters tuned with DAS (Reifenstein et al., 2024) (see Appendix S4). Colored regions represent the 95% confidence interval calculated by generating 15 instances per data point.

## 4.4    SAMPLING IN THE CASE OF BIASED DEGENERATE OPTIMAL SOLUTIONS

Next, we examine why MHCACm requires fewer steps to find any solution when there is a larger bias in the dWPE instances. Fig. 3 (a) and (b) show that the $\text{TTS}_{\text{easy}}$ for easy ground states decreases as a function of the bias $b$ of the generated Wishart instances. Conversely, the time to sampled the hard solution $\text{TTS}_{\text{hard}}$ increases. This is unlike the case of CACm without the MH step, for which the $\text{TTS}_{\text{easy}}$ and $\text{TTS}_{\text{hard}}$ remain approximately equal for all bias $b > 0$. Thus, MHCACm is better at finding the easier solutions, which explains the smaller overall TTS (to any ground state) observed in Fig. 2. Table 2 indicates that MHCACm is the fastest algorithm to sample from easier ground states, whereas CACm is the better algorithm to reach the harder to find ones.

This difference in complexity between finding the easy and hard solutions is also present in other MCMC algorithms such as SA and PT as can be seen in Tab. 2. The energy landscape contains a higher density of low-energy states near the easy solution. An algorithm that attempts to sample from a Boltzmann distribution will thus have a large bias towards this part of the solution space at finite temperature, because of the many low energy states that should be sampled with high probability. Although in the infinite time limit we would expect equal probability of finding all ground states, in practice the extremely slow relaxation time of these algorithms makes it very improbable for the "hard" ground state to be found. However, our hybrid algorithm is able to find both ground states with nonzero probability. We believe a reason for this is that the purely analog methods such as CAC, AIM and CACm do not sample from a Boltzmann distribution and thus can partially avoid these large basins of attraction. We see this in Tab. 2 in which the analog dynamics are able to find both ground states with reasonable success as well.

We show the results of the Boltzmann sampling algorithm Gibbs with gradient (Grathwohl et al., 2021) (GWG), although its purpose is not combinatorial optimization. In appendix section S6, comparison to another recently proposed sampling algorithm (Sun et al., 2023) is shown.

We also investigate the effect of changing the complexity parameter $\alpha_{\text{WPE}}$ of biased Wishart planted instances (see Fig. 4) with a bias of $b = 12$. We observe that the relative reduction of the TTS due to the introduction of the MH step in biased instances is more pronounced for instances of higher complexity (i.e., smaller parameter $\alpha_{\text{WPE}}$ as shown by the comparison of Fig. 4 (a) and (c)).

## 4.5    RESULTS ON GSET

The GSET  (Ye, 2024) is a set of Max-Cut problem instance which are commonly used to benchmark Ising and Max-Cut solvers  (Goto et al., 2021; Leleu et al., 2019; Benlic & Hao, 2013). To test the ability of MHCACm to find optimal solutions on larger optimization problems, we briefly present some results on these instances in comparison with a state of the art algorithm known as dSBM (Goto

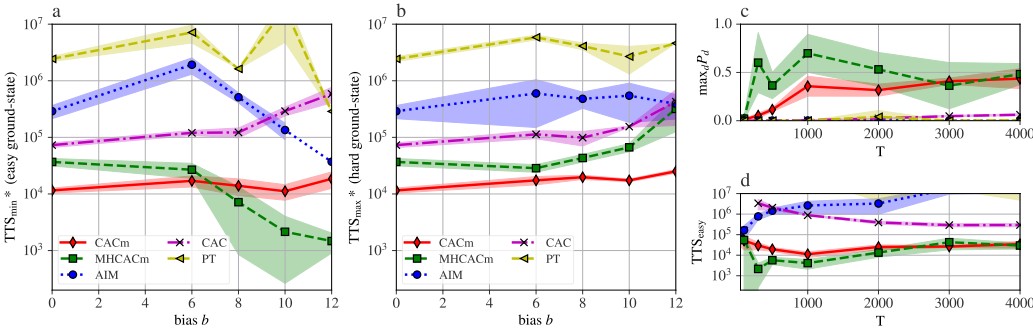

Figure 3: Time to reach the easy solution $\text{TTS}_{\text{easy}}$ (a) and hard solution $\text{TTS}_{\text{hard}}$ (b) vs. the bias $b$ of degenerate Wishart planted instances. (c) $\max_d P_d$ vs. runtime $T$ for degenerate Wishart planted instances. The same optimal parameters as in Fig. 2 are used. (d) The same for $\text{TTS}_{\text{easy}}$. (a,b,c,d) $N = 100$. (c,d) $b = 10$.

Table 2: Time to solution (in steps, i.e., matrix vector multiplication) using degenerate Wishart planted instances with bias $b = 12$. We show both TTS, $\text{TTS}_{\text{easy}}$ and $\text{TTS}_{\text{hard}}$ (as defined in the text) obtained for a variety of algorithms and two problem sizes. Since SA and GWG were not able to sample the hard ground state, $\text{TTS}_{\text{hard}}$ is not defined for it and $\text{TTS}_{\text{hard}} = \text{TTS}$. Bold numbers denote the best algorithm.

| model/N | $10^{-3}$ TTS | | $10^{-3}$ $\text{TTS}_{\text{easy}}$ | | $10^{-3}$ $\text{TTS}_{\text{hard}}$ | |
|---|---|---|---|---|---|---|
| | 100 | 140 | 100 | 140 | 100 | 140 |
| SA | 27.56 | 1917.67 | 27.56 | 1917.67 | N/A | N/A |
| PT | 2437.92 | N/A | 287.71 | N/A | 4605.06 | N/A |
| GWG | 6369.48 | 15779.94 | 6369.48 | 15779.94 | N/A | N/A |
| AIM | 22.31 | 906.96 | 37.06 | 958.13 | 390.72 | 16884.93 |
| CAC | 238.81 | 5833.26 | 586.26 | 46960.90 | 427.34 | 6634.85 |
| CACm | 9.41 | 22.91 | 18.54 | 37.84 | **25.08** | **76.88** |
| MHCACm | **1.40** | **2.44** | **1.47** | **2.44** | 318.16 | 248.63 |

et al., 2021). Table 3 shows that MHCACm reaches best known solutions faster than dSBM (Goto et al., 2021) on the first few instances of the GSET problem set (Ye, 2024).

Table 3: TTS (in steps or MVM) for solving GSET (Ye, 2024) instances of MHCACm and dSBM (Goto et al., 2021).

| instance | MHCACm | | | dSBM | | |
|---|---|---|---|---|---|---|
| $N$ (id) | $p_0$ | T | TTS | $p_0$ | T | TTS |
| 800 (1) | 0.612000 | 3000 | **14592** | 0.026777 | 2000 | 339332 |
| 800 (2) | 0.021000 | 3000 | **650949** | 0.010660 | 6000 | 2578124 |
| 800 (3) | 0.533500 | 3000 | **18118** | 0.033920 | 3000 | 400343 |
| 800 (4) | 0.140500 | 3000 | **91249** | 0.025144 | 2000 | 361673 |
| 800 (5) | 0.044000 | 3000 | **307029** | 0.022099 | 3000 | 618221 |
| 800 (6) | 0.070500 | 3000 | **188972** | 0.023856 | 1000 | 190728 |
| 800 (7) | 0.300000 | 3000 | **38734** | 0.022552 | 1000 | 201889 |
| 800 (8) | 0.166500 | 3000 | **75858** | 0.019060 | 1500 | 358947 |

## 4.6 PARALLELIZATION ON GPU

The algorithm is particularly well-suited for deployment on highly parallel hardware, such as GPUs, due to its capacity for parallel spin updates. In particular, the computational bottleneck of MHCACm is the matrix vector multiplication (MVM), a basic linear algebra operation that can be done very efficiently on modern GPUs using frameworks such as PyTorch. In table 4 we compare the wall-clock TTS of our PyTorch implementation of MHCACm finding ground states of a degenerate WPE instance of size $N = 100$. We also show the wall-clock time of a standard implementation of SA (Tiosue,

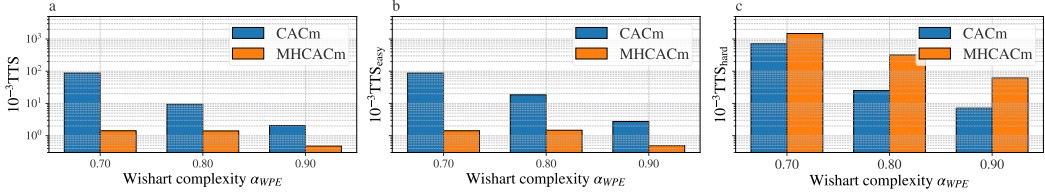

Figure 4: TTS, TTS$_{\text{easy}}$ and TTS$_{\text{hard}}$ (as defined in the text) vs. complexity of Wishart planted instances $\alpha_{\text{WPE}}$ (see definition in Appendix section S5). $N = 100$, $b = 12$.

2024) running an CPU. MHCACm on GPU exhibits a 100x speed up in real time against simulated annealing on CPU for sampling easy optimal solutions. Furthermore, MHCACm is compatible with an implementation on special-purpose analog hardware designed for linear algebra acceleration, such as a memresistor crossbar (Hu et al., 2014) or on-chip photonic mesh arrays (Bogaerts et al., 2020).

Table 4: Wall clock times of algorithms on different computing platforms. TTS is time to find any ground state of a degenerate WPE instance with $\alpha_{\text{WPE}} = 0.8$, $N = 100$ and bias $b = 12$. The GPU used is a Nvidia V100 and the CPU wall time is calculated on a 2024 MacBook Pro (Apple M3 Chip).

|  | TTS (s) ($N = 100$) | |
| --- | --- | --- |
|  | unbiased WPE | biased WPE ($b = 12$) |
| SA (CPU) | 0.0756 | 0.0567 |
| AIM (CPU) | 0.32 | 0.0287 |
| CACm (CPU) | 0.013 | 0.016 |
| MHCACm (CPU) | 0.0163 | $4.08 \times 10^{-4}$ |
| MHCACm (GPU) | **0.00207** | $\mathbf{5.18 \times 10^{-5}}$ |

## 5 COMPARISON TO OTHER APPROACHES

Methods such as HMC (Neal et al., 2011; Xifara et al., 2014) introduce auxiliary degrees of freedom (momentum) to the gradient dynamics, providing the system additional directions to move away from local minima of the error function. However, these methods do not force the dynamics to be close to the discrete state, making them ill-suited for discrete state sampling. CAC suggests a different mechanism for the auxiliary variables. It adjusts the amplitude of the continuous variables, ensuring proximity to the discrete (binary) state (Leleu et al., 2019; 2021). Conventionally, this can be interpreted as a dual-primal Lagrangian problem in which we want to minimize the error function (in the continuous embedding) while also satisfying a constraint (being close to the discrete state). Our method differs from the primal dual (Vadlamani et al., 2020) approach. With CAC, the error function is achieved by multiplying the gradient with auxiliary variables, akin to dynamically pre-conditioning the Hessian of the cost function.

Unlike HMC however, CAC does not conserve (even approximately) energy. In HMC (Neal et al., 2011; Xifara et al., 2014), the MH step compensates for the discretization errors brought about by the leapfrog integration. However, the reason for adding an MH step to CAC is different. State-of-the-art algorithms that exploit relaxation to an continuous state for discrete combinatorial optimization such as memcomputing (Sheldon et al., 2019) chaotic amplitude control (Leleu et al., 2019), and simulated bifurcation machine (Goto et al., 2021) do not accurately sample from the Boltzmann distribution. Our proposed methodology bridges this gap, integrating continuous relaxation algorithms with fair sampling of a discrete distribution via the construction of a hybrid MCMC.

## 6 CONCLUSION

The MHCACm algorithm effectively combines chaotic dynamics with the Metropolis-Hastings method to enhance ground-state sampling efficiency in discrete spaces. The introduction of the MH step increases the probability of finding more accessible ground states during non-equilibrium dynamics when a bias exists in the energy landscape. However, the MH step does not seem beneficial when such a bias among degenerate ground states is absent. This approach significantly improves the

speed and accuracy of finding optimal solutions for combinatorial optimization tasks, demonstrating superior performance and suitability for large-scale parallel processing on GPUs.

The theoretical understanding of the sampling capabilities of MHCACm are out of the scope of this paper. However, numerical evidence seems to indicate that the algorithm not only achieves fair sampling from the Boltzmann distribution after sufficient relaxation time (see Appendix Fig. S2) but can exhibit faster relaxation times on NP-hard Ising problem instances with rugged energy landscape (see Fig. 3 and Tab. 1) when compared to traditional methods such as SA. Lastly, the connection with learning in over-parameterized neural networks is to be investigated. Finding the most reachable and common solution to a large scale optimization problem is particularly useful for training neural networks in machine learning, as it typically corresponds to a trained network with better generalization properties (Soudry et al., 2018; Baity-Jesi et al., 2018; Feng & Tu, 2021; Baldassi et al., 2022; 2023).

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

- APPENDIX -

# NON-EQUILIBRIUM DYNAMICS OF HYBRID CONTINUOUS-DISCRETE GROUND-STATE SAMPLING

## S1    ANALYSIS OF THE DETERMINISTIC PATH

In the following section, we analyze the deterministic dynamical system defined as chaotic amplitude control with momentum (CACm) in the main manuscript. For simplicity, we do not take into account the effect of momentum and set $\gamma = 0$.

### S1.1    DYNAMIC PRE-CONDITIONING OF THE HESSIAN OF $V$

The equation (1) of the main manuscript is a straightforward gradient descent of the potential $V$. However, the gradient is modulated by the auxiliary variables. In this section, we give more context about the motivation for using the auxiliary variables $e$ and explain that their modulation can be interpreted as a dynamic pre-conditioning of the Hessian of $V$ at the proximity of the discrete state $\boldsymbol{\sigma}$. First, we show in Appendix section S1.2 that the variables $\boldsymbol{u}$ in the system of eqs. (1) and (2) can be eliminated by a change of variables. Then, the Jacobian matrix $J_{xx}$ in the space $x$ defined as $J_{xx} = \{\frac{\partial \dot{x}_i}{\partial x_j}\}_{ij}$ with $\dot{x}_i = \frac{dx}{dt}$ can be expressed as follows at proximity of the fixed points of the system (see Appendix section S1.3):

$$J_{xx} = D[-\alpha f(\sqrt{a}\boldsymbol{\sigma})] - D[e \circ g(\sqrt{a}\boldsymbol{\sigma})] - D[eh(\sqrt{a}\boldsymbol{\sigma})]H(\sqrt{a}\boldsymbol{\sigma}) \qquad (S1)$$

where $D[\boldsymbol{x}]$ is a diagonal matrix with elements given by the components of the vector $\boldsymbol{x}$ and $H(\sqrt{a}\boldsymbol{\sigma})$ is the Hessian of $V$ defined as the Jacobian of $\nabla_x V$ and calculated at the point $\sqrt{a}\boldsymbol{\sigma}$. The functions $f$ and $g$ and defined in Appendix section S1.2. Equation S1 shows that the auxiliary variables $e$ modulates the Hessian $H$ on the slow time scale such that $\tilde{H}(t,\tau) = D[e(\tau)h(x_i)]H$ with $h(x) = \frac{1}{\frac{d\psi}{dx}} > 0$ for $x \in\ ]-1, 1[$ and $\psi(x) = \frac{\phi^{-1}(x)}{\beta_{\text{eff}}}$ (see lemma 1 in Appendix section S1.3).

Given that the fixed points of eqs. (1) and (2) are given as $e_i = -\alpha\sqrt{a}\frac{1}{\sigma_i\frac{\partial V}{\partial x_i}(\sqrt{a}\sigma_i)}$, we can interpret the role of $e$ as a normalization of each row of the Hessian by the inverse of the gradient in the corresponding direction (see theorem 1 in Appendix section S1.3). Because the auxiliary variables $e$ remain positive, they do not change the inertia (number of positive, negative, and zero eigenvalues) of the Hessian $H$ (see lemma 2 in Appendix section S1.4) but can change its directions and eigenvalues. Moreover, we show in the theorem 2 of in Appendix section S1.5 that this dynamical system can exhibit period cycles and chaos when the target amplitude $a$ is chosen to be sufficiently small. In summary, the role of the auxiliary variables is to act as a pre-conditioning of the Hessian and introduce entropy in the search dynamics due to its chaotic dynamics. Similar dynamics have been shown to be competitive with state-of-the-art heuristics to solve Ising problems (Leleu et al., 2019; 2021).

### S1.2    CHANGE OF VARIABLE

Consider $x_i = \phi(2\beta u_i) \implies u_i = \psi_\beta(x_i)$ with $\psi_\beta(x_i) = \frac{\phi^{-1}(x_i)}{2\beta}$.

In the case $\phi(x) = \frac{2}{1+e^{-x}} - 1$, we have $\phi^{-1}(x) = -ln(\frac{2}{x+1} - 1)$, $\psi'_\beta(x) = -\frac{1}{\beta(x+1)(x-1)}$ with $\psi'_\beta(x) = \frac{\partial \psi_\beta(x)}{\partial x}$. Moreover, $\psi'_\beta(x)$ is positive on $]-1, 1[$ and $\frac{1}{\psi'_\beta(x)} \to 0$ for $|x| \to 0$.

Because $\frac{du_i}{dt} = \psi'_\beta(x_i)\frac{dx_i}{dt}$ with $\psi'_\beta = \frac{1}{\beta}\frac{\partial\phi^{-1}(x_i)}{\partial x_i}$, the equations of motion can be rewritten by eliminating $u_i$ ($\forall i \in \{1, \cdots, N\}$) as follows:

$$\frac{dx_i}{dt} = -\alpha\frac{\psi_\beta(x_i)}{\psi'_\beta(x_i)} - \frac{e_i}{\psi'_\beta(x_i)}\nabla_x V, \qquad (S2)$$

$$\frac{de_i}{dt} = -\xi(x_i^2 - a)e_i, \qquad (S3)$$

## S1.3 Jacobian matrix at the fixed points

The fixed points $\frac{dx_i}{dt} = 0$ and $\frac{de_i}{dt} = 0$ of the dynamical system described by eqs. S2 and S3 are given as follows:

$$x_i = \sigma_i \sqrt{a}, \tag{S4}$$

$$e_i = -\frac{\alpha \psi_\beta(\sigma_i \sqrt{a})}{\frac{\partial V}{\partial x_i}(\sigma \sqrt{a})}. \tag{S5}$$

Moreover, the Jacobian matrix $J$ at the fixed points is defined as:

$$J = \begin{bmatrix} J_{xx} & J_{xe} \\ J_{ex} & J_{ee} \end{bmatrix} \tag{S6}$$

with

$$J_{xx} = D[-\alpha f(\sqrt{a}\boldsymbol{\sigma})] - D[\boldsymbol{e} \circ g(\sqrt{a}\boldsymbol{\sigma})] - D[\boldsymbol{e} \circ h(\sqrt{a}\boldsymbol{\sigma})]H(\sqrt{a}\boldsymbol{\sigma}), \tag{S7}$$

$$J_{xe} = -D[\{\frac{\frac{\partial V}{\partial x_i}(\sqrt{a}\sigma_i)}{\psi'_\beta(\sqrt{a}\sigma_i)}\}_i], \tag{S8}$$

$$J_{ex} = -2\xi\sqrt{a}D[\boldsymbol{\sigma} \circ \boldsymbol{e}], \tag{S9}$$

$$J_{ee} = 0, \tag{S10}$$

with $H(x) = J[\nabla_x V]$ is the Hessian of $V$ in $x$ without its diagonal elements. The notation $D[\boldsymbol{x}]$ denote the diagonal matrix with diagonal elements given by the components of the vector $\boldsymbol{x}$. The functions $f$, $g$, and $h$ of eq. S7 are defined as follows:

$$f(x_i) = \frac{\partial \frac{\psi_\beta}{\psi'_\beta(x_i)}}{\partial x_i}, \tag{S11}$$

$$g(x_i) = \frac{\partial \frac{1}{\psi'_\beta(x_i)} \frac{\partial V}{\partial x_i}}{\partial x_i}, \tag{S12}$$

$$h(x_i) = \frac{1}{\psi'_\beta(x_i)} > 0 \text{ for } x_i \in ]-1, 1[. \tag{S13}$$

In the case of $\phi(x) = \frac{2}{1+e^{-x}} - 1$, we have:

$$f(x_i) = 2 + x_i \ln(-\frac{-1 + x_i}{1 + x_i}), \tag{S14}$$

$$g(x_i) = -\beta x_i \frac{\partial V}{\partial x_i} - \beta(x_i + 1)(x_i - 1)\frac{\partial^2 V}{\partial x_i^2}, \tag{S15}$$

$$h(x_i) = -\beta(x_i + 1)(x_i - 1). \tag{S16}$$

Note that $f$ is an even function with $f(x) = f(-x)$.

In the case $\xi \to 0$, the dynamics of $e_i$ variables becomes much slower than that of $x_i$. We note $\tau$ the slower time scale of $e_i$ with $\tau = t\xi$. The lemma 1 is obtained using eq. S7.

**Lemma 1**: When $\xi \to 0$, the Hessian $H$ of $V$ is modulated on a slow time scale $\tau$ with $\tilde{H} = D[\boldsymbol{e} \circ g(\sqrt{a}\boldsymbol{\sigma})]H$ as proximity of the fixed points $\boldsymbol{x} = \sqrt{a}\boldsymbol{\sigma}$. We call $\tilde{H}$ the effective Hessian matrix.

## S1.4 Effective Hessian

Stability of the fixed points is determined by the eigenvalues of the Jacobian matrix $J$. The blocks $J_{ex}$ and $J_{xe}$ are such that (see also (Leleu et al., 2019)):

$$J_{ex}J_{xe} = bI, \tag{S17}$$

with $I$ the identity matrix and $b = -2\xi\alpha\sqrt{a}\frac{\psi_\beta(\sqrt{a})}{\psi'_\beta(\sqrt{a})}$ and $J_{ee} = 0$. Consequently, the determinant $P(\lambda) = |J - \lambda I|$ characterizing the eigenvalues of $J$ has the following property:

$$P(\lambda) = \begin{vmatrix} J_{xx} - \lambda I & J_{xe} \\ J_{ex} & -\lambda I \end{vmatrix} = |J_{xx} - \lambda I|\lambda - J_{xe}J_{ex} = |J_{xx} - \lambda I|\lambda - bI. \tag{S18}$$

If we note $\mu_i$ the eigenvalues of $J_{xx}$, the characteristic polynomial can be rewritten:

$$P(\lambda) = -(\mu - \lambda)\lambda - bI = \lambda^2 - \mu\lambda - b. \tag{S19}$$

The eigenvalues $\lambda_j$ can come in pairs $\lambda_j^+$ and $\lambda_j^-$ depending on the sign of $\Delta_j = \mu_j^2 + 4b$ and are expressed as follows by solving $P(\lambda) = 0$ ($\forall j \in \{1, \cdots, N\}$):

$$\lambda_j^\pm = (\mu_j \pm \sqrt{\Delta_j})\frac{1}{2} \text{ if } \Delta_j > 0, \tag{S20}$$

$$\lambda_j^\pm = (\mu_j \pm i\sqrt{\Delta_j})\frac{1}{2} \text{ if } \Delta_j < 0. \tag{S21}$$

We define the pre-conditioned Hessian matrix $\tilde{\tilde{H}}$ as $\tilde{\tilde{H}}(\sqrt{a}\boldsymbol{\sigma}) = D[\frac{\sigma_i}{\frac{\partial V}{\partial x_i}(\sigma_i\sqrt{a})}]H(\sqrt{a}\sigma_i)$ such that :

$$D[e_i h(\sqrt{a}\sigma_i)]H(\sqrt{a}\boldsymbol{\sigma}) = D[-\alpha\frac{1}{\frac{\partial V}{\partial x_i}(\sigma_i\sqrt{a})}\frac{\psi_\beta(\sigma_i\sqrt{a})}{\psi'_\beta(\sigma_i\sqrt{a})}]H(\sqrt{a}\boldsymbol{\sigma}), \tag{S22}$$

$$= -\alpha\frac{\psi_\beta(\sqrt{a})}{\psi'_\beta(\sqrt{a})}D[\frac{\sigma_i}{\frac{\partial V}{\partial x_i}(\sigma_i\sqrt{a})}]H(\sqrt{a}\boldsymbol{\sigma}), \tag{S23}$$

$$= -\alpha\frac{\psi_\beta(\sqrt{a})}{\psi'_\beta(\sqrt{a})}\tilde{\tilde{H}}(\sqrt{a}\boldsymbol{\sigma}), \tag{S24}$$

The last step uses the fact that $\frac{\psi_\beta(x)}{\psi'_\beta(x)}$ is an odd function.

Note that the factors $\frac{\sigma_i}{\frac{\partial V}{\partial x_i}(\sigma_i\sqrt{a})}$, $\forall i$, multiplying the Hessian $H$ are positive at local minima satisfying $\frac{\partial V}{\partial x_i}(\sigma_i\sqrt{a})\sigma_i > 0$, $\forall i$. In this case, the eigenvalues of $D[\frac{\sigma_i}{\frac{\partial V}{\partial x_i}(\sigma_i\sqrt{a})}]H(\sqrt{a}\sigma_i)$ are the same as $D[\frac{\sigma_i}{\frac{\partial V}{\partial x_i}(\sigma_i\sqrt{a})}]^{1/2}H(\sqrt{a}\sigma_i)D[\frac{\sigma_i}{\frac{\partial V}{\partial x_i}(\sigma_i\sqrt{a})}]^{1/2}$. By Sylvester's law of matrix inertia, this implies that the eigenvalues of $\tilde{\tilde{H}}$ and $H$ have the same signs. In matrix theory, the term "inertia" refers to the number of positive, negative, and zero eigenvalues of a real symmetric matrix.

**Lemma 2**: The signs of the eigenvalues of $\tilde{\tilde{H}}$ and $H$ are the same when $\sigma_i\frac{\partial V}{\partial x_i}(\sigma_i\sqrt{a}) > 0$, $\forall i$.

**Theorem 1**: The effect of error variables at proximity of fixed points is to replace the Hessian $H$ of $V$ by an effective Hessian $\tilde{\tilde{H}}$ with $\tilde{\tilde{H}}(\sqrt{a}\boldsymbol{\sigma}) = D[\frac{\sigma_i}{\frac{\partial V}{\partial x_i}(\sigma_i\sqrt{a})}]H(\sqrt{a}\sigma_i)$. The inertia of the Hessian and effective Hessian are the same for local minima of $V$ verifying the condition $\sigma_i\frac{\partial V}{\partial x_i}(\sigma_i\sqrt{a}) > 0$, $\forall i$.

## S1.5 Eigenvalues of $H$ and stability of the fixed points

The eigenvalues of the Jacobian $J_{xx}$, noted $\mu_j$, can be expressed using the eigen spectrum of the pre-conditioned Hessian $\tilde{H}$, noted $\gamma_j$, as follows using eq. S7:

$$\mu_j = -\alpha f(\sqrt{a}\sigma_i) - e_i g(\sqrt{a}\sigma_i) + \alpha \frac{\psi_\beta(\sqrt{a})}{\psi'_\beta(\sqrt{a})}\gamma_j. \tag{S25}$$

The fixed points become stable when the real parts of eigenvalues are all negative, which is verified when the eigenvalue with maximal real part becomes equal to 0 given by the following condition using eqs. S20 and S21:

$$\max_j(\mathrm{Re}[\mu_j \pm \sqrt{\mu_j^2 + 4b}]) = 0 \text{ if } \mu_j^2 - 4b > 0, \tag{S26}$$

$$\max_j(\mathrm{Re}[\mu_j \pm i\sqrt{-\mu_j^2 - 4b}]) = 0 \text{ if } \mu_j^2 - 4b < 0. \tag{S27}$$

Noting $\tilde{\gamma}$ the eigenvalue with maximal part $\mathrm{Re}[\gamma_j]$, eqs. S26 and S26 imply that the fixed point $\sigma_i\sqrt{a}$ becomes stable under the condition (see eq. S21):

$$\tilde{\mu} = 0, \tag{S28}$$

$$\text{i.e., } -\alpha f(\sqrt{a}\sigma_i) - e_i g(\sqrt{a}\sigma_i) - \tilde{\gamma} = 0. \tag{S29}$$

Assuming $\frac{\partial^2 V}{\partial x_i^2} = 0$, we have (see eqs. S3 and S12):

$$e_i g(x_i) = \frac{\psi''_\beta(x_i)}{(\psi'_\beta(x_i))^2}\frac{\partial V}{\partial x_i}\frac{\alpha\psi_\beta(x_i)}{\frac{\partial V}{\partial x_i}}, \tag{S30}$$

$$= \alpha\frac{\psi''_\beta(x_i)}{(\psi'_\beta(x_i))^2}\psi_\beta(x_i). \tag{S31}$$

Moreover, we have (see eq. S11):

$$f(x_i) = \frac{\partial \frac{\psi_\beta}{\psi'_\beta(x_i)}}{\partial x_i} \tag{S32}$$

$$= 2 - \frac{\psi_\beta\psi''_\beta}{(\psi'_\beta(x_i))^2}. \tag{S33}$$

Combining eqs. S31 and S33 , eq. S29 can be written as follows:

$$\tilde{\gamma}(\sqrt{a}\boldsymbol{\sigma}) = 2\frac{\psi'_\beta(\sqrt{a})}{\psi_\beta(\sqrt{a})}. \tag{S34}$$

**Theorem 2**: For small values of the target amplitude $a$ such that $\tilde{\gamma}(\sqrt{a}\boldsymbol{\sigma}) > 2\frac{\psi'_\beta(\sqrt{a})}{\psi_\beta(\sqrt{a})}$ where $\tilde{\gamma}(\sqrt{a}\boldsymbol{\sigma})$ is the eigenvalue of $\tilde{H}(\sqrt{a}\boldsymbol{\sigma})$ with $\tilde{H}(\sqrt{a}\boldsymbol{\sigma}) = D[\frac{\sigma_i}{\frac{\partial V}{\partial x_i}(\sigma_i\sqrt{a})}]H(\sqrt{a}\sigma_i)$, the system described by eqs. S2 and S3 does not have any stable fixed points assuming $\frac{\partial^2 V}{\partial x_i^2} = 0$.

In general settings where the system is bounded, Theorem 2 implies that the dynamical system of eqs. S2 and S3 settles to either a periodic (quasi periodic) or chaotic attractor for sufficiently large runtime T.

In the case of the Ising Hamiltonian $V(\boldsymbol{x}) = -\frac{1}{2}\sum_{ij}\omega_{ij}x_jx_i$, the calculations are simplified because $\frac{\partial V}{\partial x_i}(\sigma_i\sqrt{a}) = \sqrt{a}\frac{\partial V}{\partial x_i}(\sigma_i)$ and $V(\boldsymbol{\sigma}\sqrt{a}) = aV(\boldsymbol{\sigma})$ and eq. S34 can be written:

$$\tilde{\gamma}(\boldsymbol{\sigma}) = 2\sqrt{a}\frac{\psi'_\beta(\sqrt{a})}{\psi_\beta(\sqrt{a})}, \tag{S35}$$

where $\tilde{\gamma}$ is independent of $\sqrt{a}$.

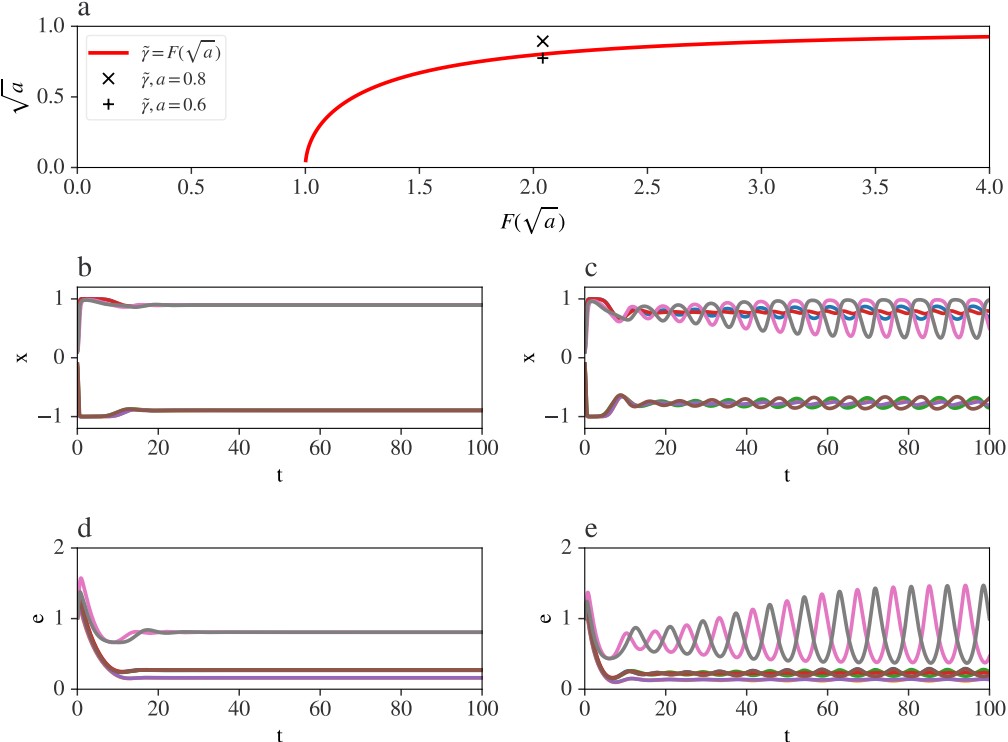

Figure S1: (a) Bifurcation diagram of the system of eqs. S2 and S3 in the case of the $V(\boldsymbol{x}) = -\frac{1}{2}\sum_{ij}\omega_{ij}\sigma_i\sigma_j$ with $N = 8$ and $\omega_{ij}$ is randomly chosen to be equal to 1 or $-1$ for $j > i$. $\omega_{ij} = \omega_{ji}$ and $\omega_{ii} = 0$. The red curve corresponds to the set $\{\sqrt{a}, F(\sqrt{a})\}$ with $a > 0$ which denote the condition under which the real part of the eigenvalues $\mu$ of the Jacobian matrix $J_{xx}$ is zero. The $+$ and $\times$ markers correspond to the maximum real part $\tilde{\gamma}$ of the eigenvalues of $D[\frac{\sigma_i}{\frac{\partial V}{\partial x_i}(\sigma_i\sqrt{a})}]H(\sqrt{a}\sigma_i)$ for the x-axis and the value of $a = 0.6$ and $a = 0.8$, respectively. (b) and (c) Time series of soft spins $x_i$ vs. time in the case $a = 0.8$ and $a = 0.6$, respectively. (d) and (e) The same as (b) and (c) for the error variables $e_i$.

In Fig. S1, the stability of the dynamics described in eqs. S2 and S3 when $V$ is Ising Hamiltonian with $N = 8$ is shown by the bifurcation diagram in the space $\{F(\sqrt{a}), \sqrt{a}\}$. The $+$ and $\times$ markers correspond to the position real part of the eigenvalue of $\tilde{\gamma}$ of $D[\frac{\sigma_i}{\frac{\partial V}{\partial x_i}(\sigma_i\sqrt{a})}]H(\sqrt{a}\sigma_i)$ for the x-axis

and the value of $a = 0.6$ and $a = 0.8$, respectively. For $a = 0.8$, the point $\{\tilde{\gamma}, \sqrt{a}\}$ is above the set defined by $\{\sqrt{a}, F(\sqrt{a})\}$ with $a > 0$ (shown in red). In this regime, the system settles to a stable fixed point as shown in Fig. S1 (b) and (d). For $a = 0.6$, the point $\{\tilde{\gamma}, \sqrt{a}\}$ is below the red curve. In this regime, the system does not possess any fixed points and the limit behavior is a periodic cycle.

## S2 PSEUDO-CODE OF MHCACM

The Pseudo-code of Metropolis-Hastings adjusted chaotic amplitude control with momentum (MH-CACm) is shown in this section.

---

**Algorithm 1** Metropolis-Hastings adjusted chaotic amplitude control with momentum

---

1: **function** DETERMINISTIC PATH$(\sigma)$      ▷ Iterate deterministic n steps of CACm
2:     $u \leftarrow 0$
3:     $u' \leftarrow 0$
4:     $u'' \leftarrow 0$
5:     $e \leftarrow 1$
6:     $y \leftarrow \sigma$      ▷ Initialize
7:     **for** $t \leftarrow 0$ to $n$ **do**
8:        $u'' \leftarrow u'$
9:        $u' \leftarrow u$
10:       $u \leftarrow u' - \alpha u' + e \circ \nabla V_x(y) + \gamma(u' - u'')$      ▷ Euler steps
11:       $e \leftarrow e - \xi(y \circ y - a) \circ e$
12:       $e \leftarrow \frac{e}{<e>}$      ▷ Normalize $e$
13:       $y \leftarrow \frac{2}{1+e^{-2\beta_{\text{eff}}u}} - 1$
14:     **end for**
15:     $y \leftarrow \frac{2}{1+e^{-2\beta_{\text{eff}}u/(\alpha e)}} - 1$
16:     **return** $y$
17: **end function**
18: **function** PROBABILISTIC JUMP$(y)$      ▷ Generate random jump
19:     $\tau = 1$ with probability $\frac{y-1}{2}$, $\tau = -1$ otherwise
20:     **return** $\tau$
21: **end function**
22: **function** METROPOLIS-HASTINGS STEP$(\tau, \sigma, y, \tilde{x})$      ▷ Metropolis-Hastings step
23:     $A(\tau/\sigma) \leftarrow \min(1, \frac{P(\tau)Q(\sigma/\tau)}{P(\sigma)Q(\tau/\sigma)}) = F(\tau, \sigma, y, \tilde{x})$      ▷ Function $F$ defined in eq. (9)
24:     $B = $ TRUE with probability $A(\tau/\sigma)$, $B = $ FALSE otherwise
25:     **if** $B$ **then**
26:       $y \leftarrow \tilde{x}$      ▷ Accept the new configuration $\tau$
27:       $\sigma \leftarrow \tau$
28:     **end if**
29:     **return** $\tau, y$
30: **end function**
31: **procedure** SAMPLE $(V)$
32:     $y \leftarrow \mathcal{U}(0,1)$      ▷ Random initialization
33:     $\sigma \leftarrow$ PROBABILISTIC JUMP $(y)$      ▷ Initial configuration
34:     **for** $k \leftarrow 0$ to $K$ **do**      ▷ Generate $K$ samples
35:       $\tau \leftarrow$ PROBABILISTIC JUMP $(y)$      ▷ New candidate configuration $\tau$
36:       $\tilde{x} \leftarrow$ DETERMINISTIC PATH $(\tau)$
37:       $\sigma, y \leftarrow$ METROPOLIS-HASTINGS STEP $(\tau, \sigma, y, \tilde{x})$
38:     **end for**
39: **end procedure**

---

## S3 Boltzmann sampling

### S3.1 Verification of Boltzmann sampling at equilibrium

To test our Markov chain approach, we sample from the Boltzmann distribution of the Ising Hamiltonian $V(\boldsymbol{\sigma}) = -\frac{1}{2}\sum_{ij}\omega_{ij}\sigma_i\sigma_j$. We use the Wishart planted ensemble (WPE) (Hamze et al., 2020) of size $N = 18$ spins to generate the Ising coupling matrix. Using the WPE to generate instances of the Ising Hamiltonian allows us to tune the complexity of the energy landscape and also know the ground state a-priori (Hamze et al., 2020). Additionally, these instances are known to exhibit properties of NP-Hard problems making the ground state difficult to find by heuristic algorithms (Hamze et al., 2020). Because we use a small problem size ($N = 18$) the exact Boltzmann distribution can be determined by brute force. The sampled distribution obtained by MHCACm is compared to the exact distribution at inverse temperature $\beta = 0.07$ by calculating the symmetric KL divergence (see Appendix section S3.2). Note that CACm (without the Metropolis-Hastings step) does not sample from the target Boltzmann distribution (see Fig. S2 a). In contrast, MHCACm exhibits exactly the same distribution of energy as the Boltzmann distribution at thermal equilibrium. This is accomplished while retaining the global exploration properties of CACm, making it a potent hybrid heuristic for identifying lower energy states of non-convex energy functions.

In order to evaluate the sampling characteristics of the constructed Markov chain, we aim at drawing samples from the Boltzmann distribution of the Ising Hamiltonian $V(\boldsymbol{\sigma}) = -\frac{1}{2}\sum_{ij}\omega_{ij}\sigma_i\sigma_j$ for a Wishart planted instance (Hamze et al., 2020) with a size of $N = 18$ spins at an inverse temperature of $\beta$.

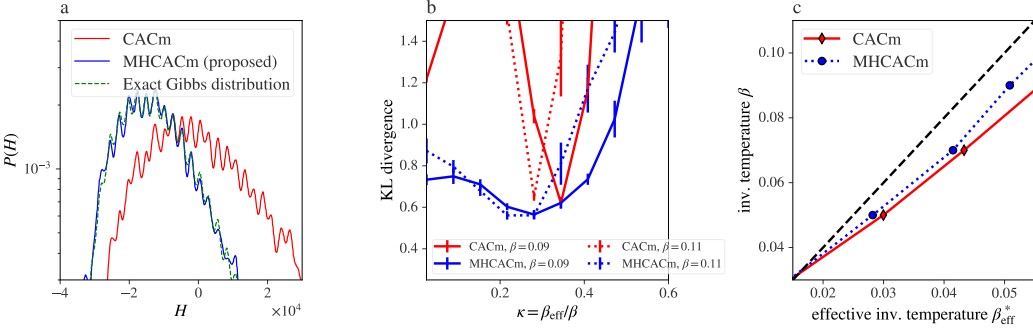

Figure S2: (a) Exact Boltzmann distribution (green dotted line), sampled distribution using CACm (without MH step, see red line), and MHCACm (without MH step, see blue line). Gaussian filtering of the energy distribution is applied for the sake of readability. $\beta = 0.07$. (b) KL divergence vs. ratio of inverse temperatures $\kappa = \frac{\beta_{\text{eff}}}{\beta}$. (c) Effective inverse temperature $\beta_{\text{eff}}^*$ temperature at which the KL divergence is minimized vs. $\beta$. The states used for the sampled distribution are boolean configurations accepted by the MH criterion.

In Figs. S2 (b) is shown that the KL divergence is minimized for a ratio of inverse temperatures $\kappa = \frac{\beta_{\text{eff}}}{\beta}$. Note that $\beta_{\text{eff}}$ is the effective temperature defined in eq. 2 of the main manuscript. The effective inverse temperature $\beta_{\text{eff}}$ at which the KL divergence is minimized temperature in noted $\beta_{\text{eff}}^*$. Figure S2 (c) shows that $\beta_{\text{eff}}^*$ is a linear function of $\beta$ within the range of parameters considered. Moreover, the addition of the MH step to MHCACm increases the similarity between $\beta^*$ and $\beta$ compared to that of CACm.

### S3.2 Simulations for small $N$

In Fig. S2, the distance to the Boltzmann distribution $P(\boldsymbol{\sigma}) = \frac{e^{\beta V(\boldsymbol{\sigma})}}{Z}$ ($Z = \sum_{\boldsymbol{\sigma}} e^{\beta V(\boldsymbol{\sigma})}$) is calculated using the symmetric KL divergence. The symmetric KL divergence $D_{\text{SKL}}(P \parallel Q)$ between the distributions $P$ and $Q$ is defined as follows:

$$D_{\text{SKL}}(P \parallel Q) = \frac{1}{2}(D_{\text{KL}}(P \parallel Q) + D_{\text{KL}}(Q \parallel P)), \tag{S36}$$

where

$$D_{\text{KL}}(P \parallel Q) = \sum_i P(i) \log \frac{P(i)}{Q(i)}. \tag{S37}$$

In the case of Fig. S2, the convergence of the MCMC is measured by the KL divergence between the exact Boltzmann distribution of energy levels obtained by brute force calculations and the estimated distribution of energy levels obtained by simulation of the Metropolis-Hastings adjusted chaotic discrete space sampling. The former computation is possible because the problems size $N = 18$ is sufficiently small for the brute force search method. The instance used for Fig. 2 is a randomly generated instance from the Wishart planted set with hardness parameter $\alpha_{\text{WPE}} = 0.8$ (Perera et al., 2020).

### S3.3 Simulations for large $N$

In order to test the convergence of the MCMC in the case of larger $N$, we can utilize a more heuristic metric. Conventional Gibbs sampling algorithm is run for a sufficiently large number of steps for the chain to converge. In the case of $N = 60$ and $\alpha_{\text{WPE}} = 0.8$, numerical simulations indicate that the convergence occurs within about 1000 time step of the Gibbs sampling algorithm. To observe the convergence, we limit the characterization of the Boltzmann distribution to its first two moments: the mean energy $< H >$ with $< H > = \sum_\sigma H(\sigma)P(\sigma)$ and standard deviation of the energy $\sqrt{< (H - < H >)^2 >}$. Figure S3 shows the convergence of the mean and standard deviation of the energy under Gibbs sampling.

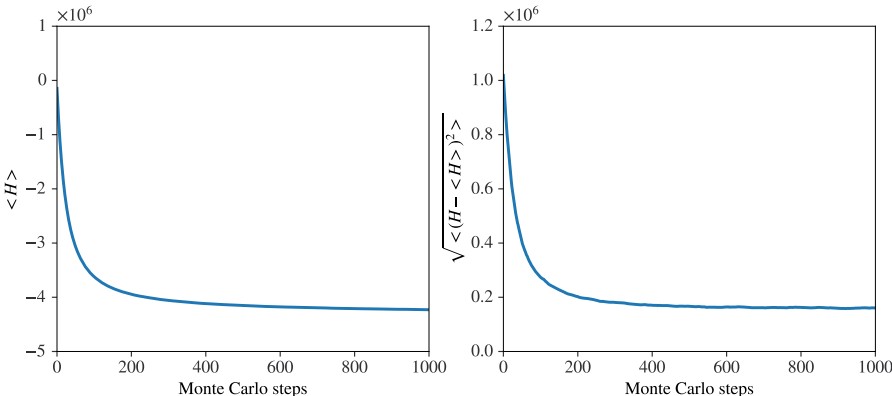

Figure S3: Convergence of the mean and standard deviation of energy distribution of a $N = 60$ Wishart planted instance with $\alpha_{\text{WPE}} = 0.8$ under Gibbs sampling. Approximate convergence is reached after 1000 Monte Carlo steps. $\beta = 2.5$.

The distance D between the mean and variance of the distribution obtained by the proposed Metropolis-Hastings adjusted chaotic discrete space sampling and the one obtained by iterating the Gibbs sampling algorithm is then calculated as follows:

$$D = | < H_{\text{chaos}} > - < H_{\text{Gibbs}} > |$$
$$+ |\sqrt{< (H_{\text{chaos}} - < H_{\text{chaos}} >)^2 >} - \sqrt{< (H_{\text{Gibbs}} - < H_{\text{Gibbs}} >)^2 >}|, \tag{S38}$$

where the subscript denote the cases of Gibbs and MH chaotic sampling, respectively.

In Fig. S4 is shown distance $D$ vs. the number of steps $T$ in the algorithm. One step corresponds to one tentative spin flip and one Euler step in the Gibbs sampling and chaotic sampling case, respectively. The proposed sampling method exhibits a faster decrease of $D$ indicating a quicker relaxation to the steady-state distribution. The use of auxiliary variables speed up the convergence compared to the case without.

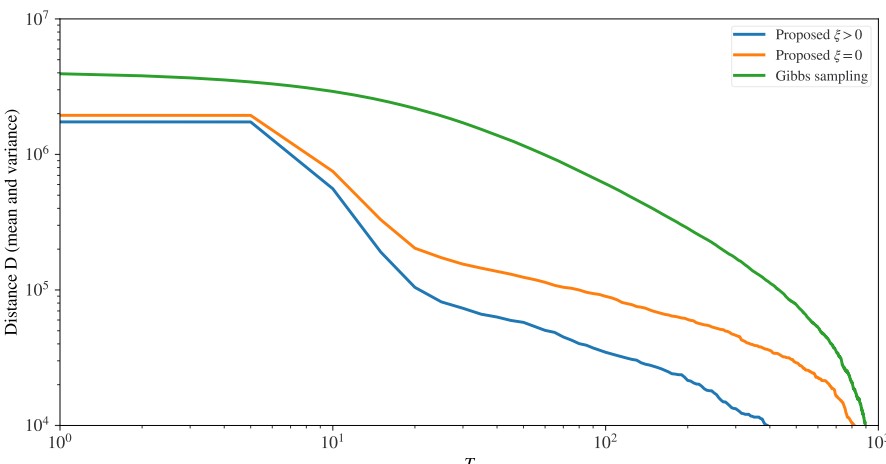

Figure S4: Distance D between the mean and variance of the distribution obtained by the proposed Metropolis-Hastings adjusted chaotic discrete space sampling and the one obtained by iterating the Gibbs sampling algorithm (see eq. S36) vs. the number of steps $T$ in the algorithm. One step corresponds to one tentative spin flip and one Euler step in the Gibbs sampling and chaotic sampling case, respectively.

For $N$ larger than $N = 60$, measuring the convergence of the distribution to its steady state becomes non-trivial because of strong broken ergodicity (Bernaschi et al., 2020). Numerical simulation of the convergence to the steady state distribution in this case falls out of the scope of this paper. Note that the convergence to the steady-state distribution can be further accelerated by considering annealing of temperature or the use of the replica exchange Monte Carlo method in both cases of Gibbs sampling and proposed chaotic sampling.

## S4  AUTOTUNING USING DYNAMIC ANISOTROPIC SMOOTHING

The parameters of the solvers are described in Table S1. We have run the autotuning algorithm called dynamic anisotropic smoothing (DAS) (Reifenstein et al., 2024) for each problem setting (problem size $N$, bias $b$). The MH sampling rate is set to $f_{\text{MH}} = 0.1$.

Table S1: Parameters of the algorithms tuned using DAS

| Algorithm | number of parameters | parameter list |
|-----------|---------------------|----------------|
| SA | 3 | $\beta_{\text{ini}}, \beta_{\text{fin}}, T$ |
| AIM | 4 | $\beta, \alpha, \gamma, T$ |
| CAC | 5 | $\beta, \alpha, \xi, a, T$ |
| CACm | 6 | $\beta, \alpha, \xi, a, \gamma, T$ |
| HMCACm | 7 | $\beta, \kappa, \alpha, \xi, a, \gamma, T$ |

An example of the the evolution of parameters during the DAS autotuning is shown in Fig. S5.

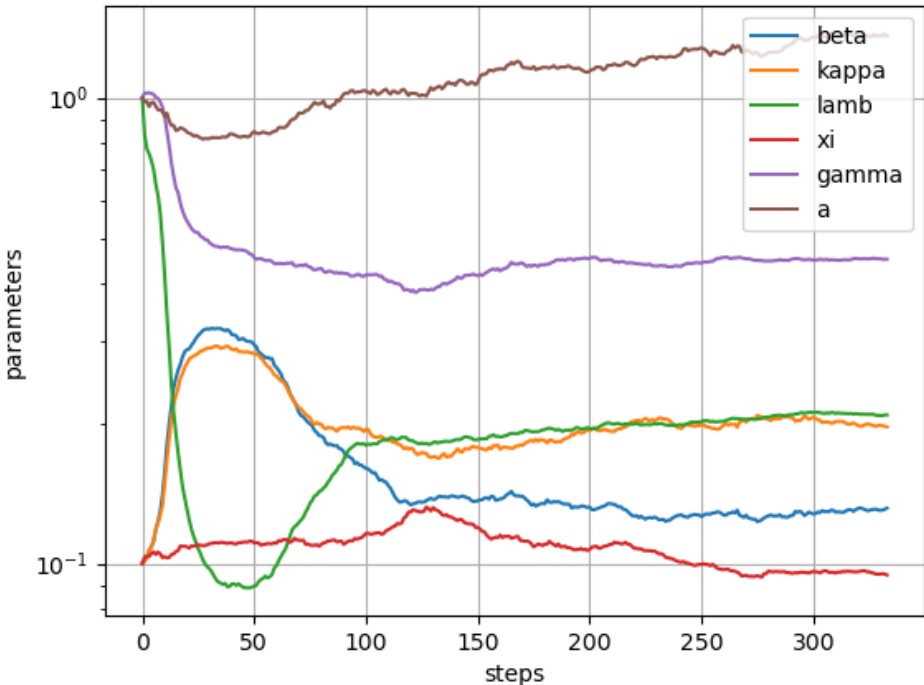

Figure S5: Evolution of parameters during the DAS autotuning for MHCACm. $N = 100$, $b = 12$, $T = 1000$. Here "lamb" denotes $\alpha$ and "kappa" is $\frac{\beta_{\text{eff}}}{\beta}$

## S5   CONTRUCTION OF DEGENERATE WISHART PLANTED ENSEMBLE

The Wishart planted ensemble instances are generated by first choosing a planted ground state $g \in \{-1, 1\}^N$. However, in our case we consider a set of planted ground states which are represented as columns in an $N$ by $D$ matrix $G \in \{-1, 1\}^{N \times D}$. Let $\hat{G}$ be a matrix with the same columns space as $G$ whose columns are orthonormal. Then we generate an $N$ by $M$ matrix $\hat{W}$ of i.i.d. Gaussian random variables. This matrix is then transformed so that the columns are orthogonal to all of the ground states to form the matrix given as follows: $W = (I - \hat{G}\hat{G}^\top)\hat{W}$. The Ising coupling matrix is constructed as $\hat{J} = WW^\top$ with the diagonal elements being set to zero. We can see that every column of $G$ is a ground state since $x^\top \hat{J} x \geq 0$ and $G^\top \hat{J} G = 0$.

Note that in the Wishart planted ensemble, the parameter $\alpha_{\text{WPE}} = \frac{M}{N}$ where $N$ is the number rows and $M$ is the number of columns, quantifies the aspect ratio of the Wishart matrix used to define the interaction strengths of the Ising model. This parameter plays a central role in determining the ruggedness of the energy landscape.

To add bias to the energy landscape, we want to somehow create the instance such that configurations close to one ground state tend to have a larger energy than the other ground state even though both ground states have the same energy. To do this, we can consider $W = (I - \hat{G}\hat{G}^\top)A\hat{W}$ where $A$ is a symmetric matrix and $\hat{W}$ is the random matrix defined above. In this work we consider $A$ to be defined as $A = I + \frac{b}{N}\mathbf{1}\mathbf{1}^\top$ where $b$ is the "bias" away from the ferromagnetic solution. The point of this is that the residual energy (energy away from the ground state) will be amplified for states that are close to the ferromagnetic solution. Then, when choosing the planted ground state we can choose one which is close to the ferromagnetic solution and one that is far creating bias in the search towards one ground state over another. For the numerical results in this work we consider one ground state with a hamming distance of 3 from the ferromagnetic solution and one ground state chosen completely randomly from $\{-1, 1\}^N$. A Gauge transform is applied to obfuscate the ground states at the end of this process.

## S6   COMPARISON TO RECENTLY PROPOSED SAMPLING ALGORITHM

In this section, we compare the proposed algorithm to the sampling method described in (Sun et al., 2023). We have implemented the algorithm of (Sun et al., 2023), that we denote by the acronym iSCO, and compare it against MHCACm in table S2. Success probability is higher using MHCACm.

|  | $p_0$ (N=60) | |
| --- | --- | --- |
|  | iSCO | MHCACm |
| b=0 (no bias) | 0.0003 | 0.0008 |
| b=12 (bias) | **0.51** | **0.15** |

Table S2: Probability $p_0$ to find any ground-state from unbiased ($b = 0$) and biased ($b = 12$) Wishart planted instances using iSCO and MHCACm after $T = 500$ time steps. $N = 60$.

## S7   COMPUTING ENVIRONMENT

All simulations are done in CPU using a Intel Core i9-11950H, 8 cores, 2.6 GHz and GPU using a NVIDIA RTX A300; except for Table 3 of the main manuscript for which the GPU used is a Nvidia V100 and the CPU wall time is calculated on a 2024 MacBook Pro (Apple M3 Chip). The code is written in python and pytorch.

