# OpenReview forum: "Non-Equilibrium Dynamics of Hybrid Continuous-Discrete Ground-State Sampling"
_ICLR.cc/2025/Conference — ICLR 2025 Poster_

### Official Review · Reviewer_SwgT · 2024-10-26

**Soundness:** 2
**Presentation:** 2
**Contribution:** 2
**Rating:** 6
**Confidence:** 3

**Summary:**

The authors propose a novel hybrid continuous-discrete algorithm that combines deterministic continuous dynamics with Metropolis-Hastings (MH) steps for ground-state sampling in non-equilibrium dynamics.

**Strengths:**

1. The combination of chaotic dynamics with an MH step to ensure convergence to a target distribution represents an innovative approach to hybrid sampling.
2. The hybrid algorithm shows potential for improving ground-state sampling performance in combinatorial optimization
3. The authors address parallelization for efficient computation on GPUs
4. The discussion on combining chaotic dynamics with probabilistic methods provides a useful context for researchers working at the intersection of machine learning and statistical physics.

**Weaknesses:**

While the authors present an interesting optimization algorithm, the clarity of the writing is a major concern. The main ideas are difficult to follow in the current presentation. I would encourage the authors to reconsider their notation and improve their writing to convey their ideas more effectively to readers.


**Major concerns**
1. The momentum and pre-conditioning typically serve different roles in optimization: momentum accumulates past gradients, and pre-conditioning captures curvature information. I do not think the connection demonstrated in Section 3.2 is trivial. A detailed explanation is needed in the main text to connect them.
2. The paper lacks theoretical guarantees regarding the convergence or performance of the proposed algorithm.
3. Since the algorithm incorporates a momentum variable, it would be more consistent to account for this momentum within the MH step (Equation (7)), rather than applying it solely to $\boldsymbol{\sigma}$.

**Other suggestions**
1. The paper lacks clear definitions for essential variables (e.g., $\boldsymbol{u}$ and $\boldsymbol{e}$ in Equation (1)).
2. The time variable $t$ is used ambiguously. It denotes continuous evolution in Equation (1) but has discrete updates in Equations (4)-(5).
3. Although the authors appear to focus on ground-state sampling, the formulation provided in Section 3.2 is more oriented toward sampling from a Gibbs measure rather than explicitly defining the ground-state sampling.
4. Ensure that all variables and abbreviations are clearly defined. For example, abbreviations such as SA, HNN, and CACm used in Table 1 should be explicitly explained in the main text.

**Questions:**

See Weaknesses.

---

> ### Author Response · Authors · 2024-11-26
>
> (C1) "While the authors present an interesting optimization algorithm, the clarity of the writing is a major concern [...]"
>
> (R1) We have made some changes to the manuscript to improve the notation and readability. In particular, we have replaced some symbols with superscripts previously used to other symbols which are easier to read.
>
> We also have rephrased several explanations. We believe the clarity of the manuscript has been substantially improved.  If you can point more specifically to notation issues, please let us know and we will change them.
>
> (C2) "The momentum and pre-conditioning typically serve different roles in optimization [...]"
>
> (R2) Momentum and pre-conditioning due to the correction of amplitude heterogeneity have indeed different roles in the dynamics. Recently proposed Ising problem solvers have used these two mechanisms independently (see AIM [Kalinin et al. 2023] and dSBM [Goto et al. 2021] for momentum and CAC [Leleu et al. 2019] for pre-conditioning related to correction of amplitude heterogeneity). Numerical experiments indeed show that the combination of the two has better performance than either one used separately. In table 2 for example, the time to solution of CACm is smaller than that of CAC and AIM.
>
> In appendix section S1, the preconditioning is analyzed without momentum. A similar analysis can be extended to take into account momentum but is out of scope of this work. In this paper, we focus on showing using numerical experiments that the combination of the two is useful for combinatorial optimization.
>
> (C3) "The paper lacks theoretical guarantees regarding the convergence or performance of the proposed algorithm"
>
> (R3) Obtaining theoretical guarantees for combinatorial optimization problems exhibiting a rugged landscape akin to spin glasses is not straightforward. It is possible to obtain using replica calculation and analytical estimation of some thermodynamic quantities, such as the number of stable fixed points etc. Even in simpler scenarios, analysis of convergence and non-equilibrium dynamics is difficult to establish (see Bernaschi 2020 in the references).
>
> Moreover, the algorithm considered in this paper exhibits asymmetric connections due to the effect of the auxiliary variables e multiplying the Ising couplings. Consequently, statistical analysis is rendered much more complicated due to the possibility of limit cycles and chaotic dynamics. This is why we have focused on numerical experiments in this work.
>
> Our current hypothesis for behavior of the algorithm is discussed in section 4.4 (see “The energy landscape is structured [...]“)
>
> (C4) "Since the algorithm incorporates a momentum variable, it would be more consistent to account for this momentum within the MH step"
>
> (R4) We completely agree, and this is indeed an interesting direction to explore in future research. In this work, we focused on a simpler scenario as an initial step—quantifying the effect of hybrid dynamics with an MH step on biased Wishart planted instances. Incorporating momentum is a natural next step in this line of investigation.
>
> (C5) "The paper lacks clear definitions for essential variables (e.g., u and e in Equation (1))."
>
> (R5) Thank you for the comment. We have added a definition for the general reader as follows:
>
> The variables x are often referred to as "soft spins." The variables u represent the internal states of these soft spins, while the auxiliary variable e accounts for variations in their amplitude.
>
> (C6) "The time variable t is used ambiguously. It denotes continuous evolution in Equation (1) but has discrete updates in Equations (4)-(5)."
>
> (R6) Thank you for pointing this out. We have replaced t by index m.
>
> (C7) "Although the authors appear to focus on ground-state sampling, the formulation provided in Section 3.2 is more oriented toward sampling from a Gibbs measure [...]"
>
> (R7)  We have clarified that we focus on ground-state sampling by explaining in section 3.2:
>
> The goal is to design a dynamical system capable of sampling from the ground states of $V$, specifically from the zero-temperature distribution P(σ) defined on the discrete space [...].
>
> (C8) "Ensure that all variables and abbreviations are clearly defined. "
>
> (R8) We have detailed the abbreviation in Table 1.

---

> > ### Comment · Reviewer_SwgT · 2024-11-26
> > **Response to the authors**
> >
> > I appreciate the authors’ response and have adjusted my score. While I recognize the potential of the proposed method, I remain concerned about the MH correction implemented in the algorithm.
> >
> > As mentioned earlier, I think it is necessary to correct both the momentum and position variables, as a detailed balance should hold for the joint distribution of these variables rather than just the marginal distribution of the position variable. Furthermore, proving that such a correction satisfies detailed balance might be a non-trivial task. I would recommend the authors carefully verify whether the proposed algorithm satisfies the detailed balance property.
> >
> > **Reference**
> >
> > A conceptual introduction to Hamiltonian Monte Carlo. arXiv:1701.02434 (2017).

---

> > > ### Author Response · Authors · 2024-11-26
> > >
> > > Thank you very much for your interest and updated score.
> > >
> > > While we totally agree with your comment in the context of HMC, we think that the situation is different in our current scenario. This is because the momentum term is effectively reset after each Metropolis-Hasting jump. We let the deterministic path run free for n steps, during which the momentum and auxiliary variables are free to be updated without MH step. Then, we set auxiliary variables e to e(nk+0)=1 and internal state u to u(nk+0) = u(nk-1) = 0 at the start of a new set of probabilistic jump and deterministic path. In our case, we do not need frequent sampling at every step, because the goal is to find degenerate ground-states in the binary configuration space rather than approximate a continuous distribution.
> > >
> > > Note that we have verified convergence to the Boltzmann distribution in Appendix section S3 and MHCACm converge in distribution very close to the Boltzmann distribution at the corresponding temperature.
> > >
> > > That said, we agree with you that it would be interesting in future work to include momentum information within the MH step without reset, making our chaotic sampling method closer to HMC. Additionally, including information about the auxiliary variables e without resetting them would be another interesting development of this approach.

---

> > > > ### Comment · Reviewer_SwgT · 2024-11-26
> > > > **Response to the authors**
> > > >
> > > > Thank you for your response. I revisited your responses, Section 3, and Algorithm 1 to gain a better understanding of the implementation.
> > > >
> > > > My primary concern remains with the correctness of the MH steps. Based on your response, I understand that $u$ and $e$ are reset after $n$ sampling steps and that the MH correction is applied every $n$ step. In other words, the implementation appears to accept $\tau$ with probability $A$ (every after $n$ step), while rejecting other variables such as $u$ and $e$ at the same time. Since $u$ and $e$ play a role in updating $\tau$, _they should also be considered in the MH correction step (accept or reject all $\tau$, $u$, and $e$ simultaneously)._
> > > >
> > > > While this approach might still perform well empirically (possibly due to the small bias introduced by ignoring $u$ and $e$ in the MH steps), _the joint distribution of all variables no longer satisfies detailed balance, and the stationary distribution may not converge to the target anymore._ In other words, since the algorithm still does not satisfy detailed balance, why do we need the extra MH steps? From my perspective, this remains a significant theoretical issue.
> > > >
> > > > Additionally, I noticed several issues with the clarity and consistency of the notations. For instance, the variable $\tau$ in Eq. (7) is not clearly defined elsewhere (maybe set $\tau=x((k+1)n)$?). In Algorithm 1 (Appendix S2), the input to "DETERMINISTIC PATH" ($\sigma$) and its output ($x$) are not explicitly linked. The input $y$ to "PROBABILISTIC JUMP" is not used, and the inputs $y$ and $\tilde{x}$ in "METROPOLIS-HASTINGS STEP" are not utilized in line 22. Furthermore, there seems to be a missing step to update $\tau$ after computing $A$ in line 22.
> > > >
> > > > I recommend revisiting the notation and definitions to improve its clarity and accessibility. Additionally, moving Algorithm 1 to the main text might help readers better follow the proposed method and understand its implementation.

---

> > > > > ### Author Response · Authors · 2024-11-27
> > > > >
> > > > > We really appreciate you taking the time to check again and, indeed, there were some clarifications that were needed; especially in the pseudo-code.
> > > > >
> > > > > The definition of $\tau$ is given in the paragraph after equation (7), at the top of page 5. We have added a more direct definition for clarify.
> > > > > We have made the pseudo-code of appendix S2 more explicit and made a few updates to the notation:
> > > > > - we have removed $p$ and used only $y$ in “DETERMINISTIC PATH” for simplicity,
> > > > > - the input to PROBABILISTIC JUMP is fixed,
> > > > > - we have made explicit that the “METROPOLIS-HASTINGS STEP” depends also on $y$ and $\tilde{x}$ ,
> > > > > - we have written explicitly the update of variables for $\tau$.
> > > > >
> > > > > We agree that our previous version of the pseudo-code was too reliant on implicit information written in the main text and we hope this version is clearer. Thank you very much for pointing this out.
> > > > >
> > > > > Concerning the dependance of the update rule on the momentum term, the terms $u$ and $e$ are solely a function of the initial state $\sigma$, given that $u$ and $e$ are reset at every deterministic path. Thus, $y$ and, in turn, $\tau$ only depends on the initial state of the deterministic trajectory $\sigma$. Numerical results support this approach in practice.
> > > > >
> > > > > We think it is better to focus on the main numerical results in the main manuscript rather than detailing the pseudo-code, which is taking too much space.

---

> ### Comment · Reviewer_SwgT · 2024-11-29
> **Response to the authors**
>
> Thanks for the authors’ detailed explanation. Upon revisiting the algorithm's MH steps, I may misunderstand the implementation. I rechecked the algorithm and the algorithm should satisfy the detailed balance. I have adjusted my score accordingly.

---

### Official Review · Reviewer_zbqu · 2024-11-03

**Soundness:** 3
**Presentation:** 3
**Contribution:** 3
**Rating:** 6
**Confidence:** 4

**Summary:**

This paper presents two variants of the CAC (chaotic amplitude control) algorithm, namely CAC with momentum (CACm) and Metropolis-Hastings CAC with momentum (HMCACm). CACm is a deterministic continuous-time dynamical model for combinatorial optimization, and HMCACm is a Metropolis-Hastings adjusted version of CACm with the Boltzmann distribution as the theoretical equilibrium. MHCACm can be regarded as a unified framework that generalizes many existing methods, including simulated annealing, Hopfield neural networks, analog iterative machines, and CAC(m). Numerical results show that this method illustrates faster relaxation time on NP-hard problems. In particular, MHCACm exhibits excellent performance in sampling from easier ground states, which may be relevant to training over-parametrized neural networks.

**Strengths:**

- State-of-the-art algorithms exploiting relaxation to a continuous state form discrete combinatorial optimization do not sample fairly from the Boltzmann distribution due to the lack of detailed balance. The MHCACm algorithm fills in this conceptual gap by adding a Metropolis-Hasting step. This new design ensures that MHCACm samples fairly from a discrete distribution while iterating over the relaxed (continuous) search space.
- The numerical results look strong. Table 3 shows that MHCACm has a success probability higher than dSBM, another well-known Ising solver based on GPU.
- MHCACm is well-suited for large-scale deployment on GPU because its computational bottleneck is matrix-vector multiplication.

**Weaknesses:**

- Little theoretical justification for the effectiveness of MHCACm is provided. Specifically, it is not clear why a fair sampling strategy in the CAC framework would lead to a better performance in an optimization (ground state finding) problem. Relaxation to the ground state does not necessarily need to go through a detailed-balance algorithmic path. It would be nice to discuss how the Metropolis-Hastings step interacts with the CAC dynamics to potentially improve optimization performance.
- No empirical results for real-world optimization problems. While the performance of MHCACm has been benchmarked over the dWPE instances and GSET, these test instances are highly artificial and may not reflect the performance of the algorithm in a practical setting (e.g., quadratic assignment problems, portfolio optimization problems, etc.).

**Questions:**

- The paper only reports the TTS in the experiments on dWPE instances (section 4.4). It would be more transparent if the success probability and runtime data could be provided as well, as it is not clear whether the advantage comes from a higher success probability or a shorter wall-clock runtime due to GPU parallelization.
- Can you elaborate more on the "dual-primal Lagrangian approach" and its difference from CAC?
- Some minor typos: e.g., line 122 "in order (to) benchmark this algorithm's ability".

---

> ### Author Response · Authors · 2024-11-26
>
> (C1) "Specifically, it is not clear why a fair sampling strategy in the CAC framework would lead to a better performance in an optimization (ground state finding) problem [...]"
>
> (R1) Our paper primarily focuses on numerical results and the observation that there is a difference in the behavior of CAC with and without the Metropolis-Hastings step. To explore this, we construct a new type of planted instances that exhibit a tunable bias within their degenerate ground states.
> While we agree that a theoretical justification would be highly interesting, the effects discussed in this paper are non-equilibrium dynamical effects, for which developing a theoretical framework can be challenging (refer to Bernaschi 2020 in the references).
> Moreover, the algorithm analyzed in this paper introduces asymmetric connections due to the influence of auxiliary variables e multiplying the Ising couplings. This asymmetry significantly complicates statistical analysis, as it creates the potential for limit cycles and chaotic dynamics. Providing a theoretical justification would require substantial additional work, which is beyond the scope of this paper. The current work already presents new ideas and concepts.
>
> Our current hypothesis for behavior of the algorithm is discussed in section 4.4 (see “The energy landscape is structured [...]“)
>
> (C2) "No empirical results for real-world optimization problems [...]"
>
> (R2) We agree with you. However, there is a limitation when working with real-world optimization problems: the ground state is not known a priori. From an experimental perspective, using planted instances is much more rigorous. It would be interesting to explore the application of our methods to real-world problems in future work.
>
> (C3) "It would be more transparent if the success probability and runtime data could be provided as well [...]"
>
> (R3) Thank you for this suggestion. We have revised Figure 3 to include subplot (c), which shows the probability of finding the "easy-to-reach" ground states, along with the corresponding TTS_easy​ in subplot (d). The additional results demonstrate that the improved performance of MHCACm is indeed due to a higher success probability, rather than GPU parallelization.
>
> (C4) "Can you elaborate more on the "dual-primal Lagrangian approach" and its difference from CAC?"
>
> (R4) The relationship between CAC and the dual-primal Lagrangian approach is discussed in [1]. CAC is based on the concept of relaxing binary variables to continuous variables (or soft spins), using gradient descent, and employing auxiliary variables to modulate the dynamics and constrain the system to return to a binary state after a transient phase.
>
> The dual-primal Lagrangian approach achieves a similar objective through the concept of descent-ascent, where gradient descent is performed in the relaxed continuous space of soft spins, and gradient ascent is carried out in the space of Lagrangian multipliers used to enforce a binary state constraint.
>
> However, [1] demonstrates that the auxiliary variables used in the dual-primal Lagrangian approach and CAC are not equivalent: in CAC, the auxiliary variables act as a pre-factor to the gradient, which is not the case in the dual-primal Lagrangian approach. As a result, CAC introduces effective asymmetric connections.
>
> [1] Sri Krishna Vadlamani, Tianyao Patrick Xiao, and Eli Yablonovitch. Physics successfully implements lagrange multiplier optimization. Proceedings of the National Academy of Sciences, 117(43): 26639–26650, 2020.
>
> (C5) "Some minor typos: e.g., line 122 'in order (to) benchmark this algorithm's ability'"
>
> (R5) Thanks.

---

> > ### Comment · Reviewer_zbqu · 2024-11-27
> >
> > We thank the authors for their response. I have decided to maintain my current score.

---

### Official Review · Reviewer_g5qM · 2024-11-04

**Soundness:** 3
**Presentation:** 3
**Contribution:** 3
**Rating:** 6
**Confidence:** 2

**Summary:**

The paper proposes a class of hybrid continuous-discrete algorithms by integrating continuous dynamics with Metropolis-Hastings steps. The paper also constructs a set of Ising problems with a tunable parameter to trade off between easy ground states and hard degenerate ground states, in order to experiment with the bias of different algorithms. The proposed class of algorithms are also fast solvers that achieve a great amount of acceleration on GPU due to a parallelizable structure.

**Strengths:**

* The paper writing is nice and structured, with a comprehensive literature review and detailed problem set-up.
* The paper looks technically sound with solid mathematical proof.
* The proposed algorithm is evaluated on multiple tasks and compared with various other benchmark methods, showing competitive performance.

**Weaknesses:**

* I am not very familiar with the literature, but seems that the tasks of ground-state sampling are not formally defined in the paper, as well as the idea of non-equilibrium dynamics.
* The connection between ground-state sampling and deep learning optimization/generalization mentioned in the paper is interesting, but the discussion is very limited.
* For numerical experiments, the definition of TTS is hard to comprehend. Does smaller TTS indicate better algorithmic performance?

**Questions:**

* Based on the algorithmic design in this paper, is there any insight we can draw on what an ideal optimizer for deep neural nets should look like?
* Can you elaborate more on the numerical performance of CACm and MHCACm? From the charts, the two performances seem to be close to each other.

---

> ### Author Response · Authors · 2024-11-25
>
> (C1) "the tasks of ground-state sampling are not formally defined in the paper, as well as the idea of non-equilibrium dynamics"
>
> (R1) Thank you for the suggestion. We have added the following explanation in introduction:
>
> “Ground-state sampling involves finding not just any ground state, as it is often defined in combinatorial optimization, but multiple degenerate ground states. In this context, non-equilibrium dynamics refers to the processes through which systems evolve over time toward steady states, with varying rates of reaching degenerate ground states that differ from equilibrium expectations”
>
> (C2) "The connection between ground-state sampling and deep learning optimization/generalization mentioned in the paper is interesting, but the discussion is very limited."
>
> (R2) The connection between ground-state sampling and deep neural networks involves the concept of implicit bias (the inherent tendencies of optimization algorithms, such as stochastic gradient descent, to prefer certain solutions or behaviors over others) and the fact that there are many solutions of zero training error in overparameterized deep neural networks (see more details in [Soudry et al., 2018; Baity-Jesi et al., 2018; Feng & Tu, 2021; Baldassi et al., 2022; 2023]. From the viewpoint of combinatorial optimization, these concepts are reminiscent of non-equilibrium dynamics and sampling of degenerate ground-states. Although this connection is important, it is the subject of future work to develop it further.
>
> (C3) "For numerical experiments, the definition of TTS is hard to comprehend. Does smaller TTS indicate better algorithmic performance?"
>
> (R3) Yes, we have added the following to be clear:
>
> A common metric for evaluating the performance of Ising solvers is the "time to solution" (TTS) which measures the number of steps needed to have 99% probability of finding any ground state (the smaller, the better the algorithm's performance).
>
> (C4) "Based on the algorithmic design in this paper, is there any insight we can draw on what an ideal optimizer for deep neural nets should look like?"
>
> (R4) When applied to learning in deep neural networks, our results suggest that intermittent jumps, coupled with Metropolis-Hastings (MH) corrections, could enhance optimization by facilitating transitions to states corresponding to larger basins of attraction. These states are often linked to better generalization, as indicated in some literature.
>
> (C5) "Can you elaborate more on the numerical performance of CACm and MHCACm? From the charts, the two performances seem to be close to each other."
>
> (R5) In the case of biased Wishart planted instances, the performance of MHCACm is about 10x better than CACm and AIM (see Fig. 3 (a) at b=12 and table 4). Indeed, the time to solution is 10x smaller for MHCACm. This is a significant difference given that AIM and CACm are state of the art algorithms for combinatorial optimization.

---

> > ### Comment · Reviewer_g5qM · 2024-12-01
> >
> > I appreciated the author's further explanations and clarifications. I am happy to maintain my score to reflect my positive support for this manuscript.

---

### Official Review · Reviewer_wvMs · 2024-11-05

**Soundness:** 3
**Presentation:** 3
**Contribution:** 3
**Rating:** 6
**Confidence:** 3

**Summary:**

Overall, this paper proposes a promising hybrid continuous-discrete sampling framework that demonstrates clear benefits in convergence speed and sampling accuracy for rugged energy landscapes. This paper could benefit from additional theoretical insights and comparisons with established methods.

**Strengths:**

(1)The proposed method effectively leverages the Metropolis-Hastings method within a continuous-discrete framework, enhancing sampling efficiency for ground-state discovery in challenging discrete landscapes. This approach demonstrates practical advantages, such as notable improvements in convergence speed and sampling accuracy on GPU architectures.

(2) The method’s focus on non-equilibrium dynamics and its capacity to identify accessible ground states faster than traditional approaches offer a valuable contribution to optimization in rugged energy landscapes.

**Weaknesses:**

(1) Although the paper demonstrates the practical benefits of the hybrid continuous-discrete approach, the theoretical understanding of the sampling properties of the MHCACm algorithm remains unaddressed. I wonder if the authors could provide a discussion on potential directions for analytical proof of the sampling capabilities of MHCACm, such as convergence rates or mixing times.

(2) I would like to suggest the authors include more comparison with other prominent sampling algorithms using collective variables, for example, 'Sampling metastable systems using collective variables and Jarzynski–Crooks paths' by G. Stoltz et al.

In particular, I am curious  to see how the use of collective variables in that work relates to or differs from the proposed method of this paper, and if the authors can combine both approaches.

**Questions:**

(1)  While the method achieves a 100x speedup over simulated annealing on GPUs, a discussion of any limitations or computational trade-offs encountered in specific scenarios (such as highly multimodal landscapes) would be beneficial, for instance, I wonder if the authors could provide specific examples of problem types or landscapes where their method may face challenges.

(2) I wonder if the authors could provide additional insights into how MHCACm scales with increased problem complexity, for instance, if the authors could demonstrate how the empirical scaling results and the performance of proposed algorithm changes with increasing complexity for a range of benchmark examples.

(3)  The paper implies that the method’s bias towards “easy” ground states is advantageous, but this effect could also limit the algorithm’s ability to reach more challenging or rare ground states. I wonder if the authors could provide quantitative results on the algorithm's performance in finding both "easy" and "hard" ground states across different problem instances.

(4) Additionally, I wonder if the authors could discuss potential modifications to the algorithm that could help balance the performance of exploration of both easy and hard ground states.

---

> ### Author Response · Authors · 2024-11-25
>
> (C1) " the theoretical understanding of the sampling properties of the MHCACm algorithm remains unaddressed [...]"
>
> (R1) Obtaining theoretical guarantees for combinatorial optimization problems with a rugged landscape resembling spin glasses is challenging. Analytical estimations of certain thermodynamic quantities, such as the number of stable fixed points and mixing times, can be derived using replica calculations. However, even in simpler cases, establishing convergence and non-equilibrium dynamics analysis remains difficult (refer to Bernaschi 2020 in the references).
>
> The algorithm analyzed in this paper introduces asymmetric connections due to the influence of auxiliary variables e multiplying the Ising couplings. This asymmetry significantly complicates statistical analysis, as it introduces the potential for limit cycles and chaotic dynamics. Although possible avenues of analysis are the dynamical cavity approach and related methods, our focus in this study has been on conducting numerical experiments.
>
> Our current hypothesis for behavior of the algorithm is discussed in section 4.4 (see “The energy landscape is structured [...]“)
>
> (C2) "I would like to suggest the authors include more comparison with other prominent sampling algorithms using collective variables, for example, 'Sampling metastable systems using collective variables and Jarzynski–Crooks paths' by G. Stoltz et al [...]"
>
> (R2) Thank you for suggesting this interesting work. We have added the reference to our manuscript to make the references more exhaustive. Since the algorithm by G. Stoltz et al. is designed for continuous sampling rather than discrete, it could not be easily applied to our benchmark.
>
> We think that combining the two approaches is not trivial, given that it is not straightforward to apply dynamic collective variables space to our scenario of discrete optimization in the binary space. Our approach uses a relaxation to continuous dynamics, but the underlying problem is discrete. Finding out how to combine these two approaches can be the subject of interesting future works.
>
> (C3) "I wonder if the authors could provide specific examples of problem types or landscapes where their method may face challenges."
>
> (R3) In the manuscript, we compare the impact of adding the MH correction in two scenarios: unbiased degenerate Wishart planted instances (where all ground states have symmetric properties in the energy landscape) and biased degenerate Wishart planted instances (where some degenerate ground states are more easily reachable due to the structure of the energy landscape).
> As shown in Fig. 2, the unbiased instances are examples for which the introduction of the hybrid approach with the MH step hurts performance of the algorithm, whereas it helps in the case of the biased instances.
> The unbiased Wishart planted instances are one example which showcases the limits of the hybrid approach.
>
> (C4) "I wonder if the authors could provide additional insights into how MHCACm scales with increased problem complexity [...]"
>
> Thank you very much for the interesting comment. We conducted additional numerical simulations and considered the impact of the parameter α_WPE, which determines the complexity of Wishart planted instances. For certain values of α_WPE, the recovery of the planted solution becomes easier or harder.
>
> We added the new Figure 4, which compares performance of CACm and MHCACm with respect to this complexity parameter α_WPE. We observe that the relative reduction in TTS due to the introduction of the MH step (for the biased WPE case) is more pronounced for instances of higher complexity (i.e., smaller parameter αWPE).
>
> (C5) "I wonder if the authors could provide quantitative results on the algorithm's performance in finding both "easy" and "hard" ground states across different problem instances"
>
> The answer to your question is contained in Figure 3, where the time to find “easy” and “hard” ground-states are compared in Fig. 3 (a) and (b), respectively. It is shown that the hybrid algorithm MHCACm shows reduced time to find “easy” ground-state for biased instances (b>>0).
>
> (C6) "Additionally, I wonder if the authors could discuss potential modifications to the algorithm that could help balance the performance of exploration of both easy and hard ground states."
>
> (R6) In the current algorithm, the acceptance criterion of the MH step does not really deal with the asymmetric flow of the dynamics due to the auxiliary variables e since the variables e are reset to 1 after each MH step. Given than CACm shows similar performance for both “easy” and “hard”, it is possible that modifying the MH step to take advantage of the information contained in the e variables could provide the benefits of MH step for both “hard” and “easy” ground-state. However, this is a speculation of this stage and further work is needed to explore this possibility.

---

> > ### Comment · Reviewer_wvMs · 2024-11-26
> >
> > Thank you for the response and all updates! I modified my rating score for this work.

---

### Official Review · Reviewer_ymkA · 2024-11-09

**Soundness:** 3
**Presentation:** 2
**Contribution:** 2
**Rating:** 6
**Confidence:** 3

**Summary:**

This paper proposes a new algorithm that combines chaotic search and Metropolis-Hastings. The goal seems to solve optimization problems in discrete non-convex energy landscapes. The proposed algorithm is tested on several combinatorial optimization tasks.

**Strengths:**

- The empirical results include multiple baselines and comparisons in terms of different metrics.
- The visualization in Fig.1 clearly shows the main algorithmic idea.

**Weaknesses:**

- The problem that this paper aims to solve is vague. Is the goal to develop an algorithm that samples better in non-convex energy landscapes, or for optimization in discrete landscapes? Or is the goal to understand non-equilibrium dynamics in non-convex energy landscapes? Similarly, the motivation of the proposed algorithm which combines chaotic search with Metropolis-Hastings is not well-explained.
- The novelty of the proposed algorithm is unclear. Is the algorithm a straightforward combination of chaotic search and MH? If not, what is the challenge, and how does the paper solve the challenge?
- The empirical improvement is not consistent. For example, Fig.2 shows that CACm is better than proposed method also the variance of the proposed is significantly larger than the baselines.
- The runtime comparison only considers simulated annealing. It will be better to include other baselines as well.
- The paper compared the standard Gibbs with gradient which is developed for combinatorial optimization. It will be more convincing to compare with gradient-based discrete MCMC that is developed for CO, such as [1].

[1] Revisiting sampling for combinatorial optimization, ICML 2023

**Questions:**

NA

---

> ### Author Response · Authors · 2024-11-25
>
> (C1) "The problem that this paper aims to solve is vague [...]"
>
> (R1) We have added a definition of the task of ground-state sampling, which is the focus of this work, in the introduction to make things clearer:
>
> “Ground-state sampling involves finding not just any ground state, as it is often defined in combinatorial optimization, but multiple degenerate ground states. In this context, non-equilibrium dynamics refers to the processes through which systems evolve over time toward steady states, with varying rates of reaching degenerate ground states that differ from equilibrium expectations“
>
> The goal is to study the effect of adding an MH step on the time required to find degenerate ground states and how this depends on the shape of the energy landscape. This is achieved by comparing biased and unbiased Wishart planted instances and other benchmark sets. A byproduct of this work is the definition of a general algorithm that, when properly tuned, demonstrates state-of-the-art performance on several benchmarks.
>
> (C2) "The novelty of the proposed algorithm is unclear [...]"
>
> (R2) The main contributions of this work are as follows:
>
> 1) A unified framework, MHCACm, that generalizes many existing methods, including simulated annealing, Hopfield neural networks, analog iterative machines, and CACm. This framework achieves optimal performance when parameters are properly tuned.
>
> 2) A new set of planted instances enables the study of the effects of non-equilibrium dynamics. We observe that the optimal algorithm (or parameter settings for a general algorithm) depends on whether the objective is to sample any ground state or all ground states.
> The combination of these two contributions enables the study of how the time to find degenerate ground states is influenced by the effects of hybridization with the MH step and various algorithmic settings.
>
> (C3) "The empirical improvement is not consistent. For example, Fig.2 shows that CACm is better than proposed method also the variance of the proposed is significantly larger than the baselines."
>
> (R3) Let us rephrase your comment to ensure we understand it correctly:
>
> 1) CACm has a lower TTS than MHCACm in Fig. 2a.
> 2) MHCACm exhibits larger variance in Fig. 2b.
>
> Our response:
>
> The main argument of the paper is that there is no single “winner” algorithm for all scenarios. Specifically, for the task of sampling from unbiased Wishart instances, CACm indeed performs better. However, in the case of biased instances, MHCACm outperforms CACm. This highlights the "no free lunch" principle.
>
> Figure 3 explains why MHCACm is better when there is a bias: MHCACm is more effective at finding “easy-to-reach” ground states (Fig. 3a) but is less effective at finding the “hard-to-reach” ones (Fig. 3b). In contrast, CACm does not exhibit significant differences between these cases. Therefore, when the task is to find any ground state, MHCACm achieves this much faster (resulting in a smaller TTS), which accounts for the difference observed in Fig. 2a.
>
> The larger variance in TTS for MHCACm, as shown in Fig. 2b, is indeed expected. This is because the optimal TTS for MHCACm occurs at smaller values of T and p0​ , which naturally increases the variance of TTS. This is supported by Fig. 3c, where the optimal T for MHCACm, corresponding to a smaller TTS, is much lower than that of CACm.
>
> (C4) "The runtime comparison only considers simulated annealing. It will be better to include other baselines as well."
>
> (R4) Thank you for the suggestion. We have included AIM and CACm on CPU as additional baselines, as these two algorithms are considered state-of-the-art. As anticipated, MHCACm outperforms the others on CPU for biased instances.
>
> (C5) "The paper compared the standard Gibbs with gradient which is developed for combinatorial optimization. It will be more convincing to compare with gradient-based discrete MCMC that is developed for CO, such as [1].
> [1] Revisiting sampling for combinatorial optimization, ICML 2023"
>
> (R5) We have experimented with the algorithm mentioned in the paper you referenced and included the numerical results in Appendix S6. MHCACm appears to exhibit a higher probability of finding ground states in both unbiased and biased instances. We would be happy to discuss this point in more detail if you are interested.

---

> > ### Comment · Reviewer_ymkA · 2024-11-26
> >
> > I thank the authors for their detailed responses, especially adding additional experiments. Most of my concerns are resolved. I have updated my score.

---

### Author Response · Authors · 2024-11-25
**Revisions**

We want to thank the referees for their insightful comments. We have run additional experiments and made some changes to the manuscript to respond to their questions. We believe the manuscript is substantially improved by these changes. The list of updates are as follows:

1) Figure 3 has been modified to show the success probability of finding ground-states (see Panel (c)).
2) A new Figure 4 has been added, which shows the dependence of the time to solution with respect to the complexity parameter α_WPE of Wishart planted instances.
3) In Table 4, the run time in seconds of additional algorithms (AIM, CACm) have been added and typos in the other numbers updated. Conclusions derived from these results are the same as in the previous version.
4) Additional numerical results in Appendix S6 for comparison with a recently proposed sampling algorithm.
5) The notation has been improved for clarity.
6) Some explanations have been rephrased to improve readability.

Major revisions to the manuscript are shown in red color.

---

### Comment · Area_Chair_32H9 · 2024-11-26

Dear Reviewers ymkA, g5qM, zbqu, SwgT,
If not already, could you please take a look at the authors' rebuttal? Thank you for this important service.
-AC

---

### Meta-Review · Area_Chair_32H9 · 2024-12-19

**Metareview:**

This paper considers designing non-equilibrium dynamics for computing the ground state of rugged energy landscapes. Two variants of the CAC (chaotic amplitude control) algorithm, namely CAC with momentum (CACm) and Metropolis-Hastings CAC with momentum (HMCACm) were proposed. Empirical results show fast convergence of the methods. Reviewers expressed concerns about the precise definition of the problem to solve, comparison with existing approaches, and theoretical justification, but after discussions they suggested most of the concerns were resolved. My impression is overall the strengths overweight the weaknesses, hence the recommendation of acceptance. However, the authors should account for the discussions in a revision.

**Additional Comments On Reviewer Discussion:**

Reviewers expressed concerns about the precise definition of the problem to solve, comparison with existing approaches, and theoretical justification, but after discussions they suggested most of the concerns were resolved. My impression is overall the strengths overweight the weaknesses, hence the recommendation of acceptance.

---

### Decision · Program_Chairs · 2025-01-22

Accept (Poster)